# How does Uncertainty-aware Sample-selection Help Decision against Action Noise?

## Abstract

Learning from imperfect demonstrations has become a vital problem in imitation learning (IL). Since the assumption of the collected demonstrations are optimal cannot always hold in real-world tasks, many previous works considers learning from a mixture of optimal and sub-optimal demonstrations. On the other hand, video records can be hands-down demonstrations in practice. Leveraging such demonstrations requires annotators to output action for each frame. However, action noise always occurs when the annotators are not domain experts, or meet confusing state frames. Previous IL methods can be vulnerable to such demonstrations with *state-dependent action noise*. To tackle this problem, we propose a robust learning paradigm called USN, which bridges Uncertainty-aware Sample-selection with Negative learning. First, IL model feeds forward all demonstration data and estimates its predictive uncertainty. Then, we select large-loss samples in the light of the uncertainty measures. Next, we update the model parameters with additional negative learning on the selected samples. Empirical results on Box2D tasks and Atari games demonstrate that USN improves the performance of state-of-the-art IL methods by more than 10% under a large portion of action noise.

## 1 Introduction

Despite the great success of reinforcement learning (RL) (Sutton & Barto, 2018) over last few years, designing hand-crafted reward functions can be extremely difficult and even impossible in many real-world tasks (Ng et al., 1999; Amodei et al., 2016; Brown et al., 2019a). Alternatively, imitation learning (IL) (Russell, 1998; Schaal, 1999; Abbeel & Ng, 2004; Argall et al., 2009; Hussein et al., 2017) aims to train an agent to mimic the demonstrations collected from an expert, without any access to hand-crafted reward signals. However, it is expensive and difficult to collect high-quality demonstrations in real-world tasks (Silver et al., 2013).

In practice, it is much cheaper to collect demonstrations from amateurs (Audiffren et al., 2015). Existing works (Tangkaratt et al., 2019; 2020; Zhang et al., 2021b) have studied imitation learning from a mixture of optimal and non-optimal demonstrations. Specifically, Tangkaratt et al. (2019) requires that all the actions for a demonstration are drawn from the same noisy distribution with sufficiently small variance. Following works (Tangkaratt et al., 2020) proposed robust imitation learning by optimizing a classification risk with a symmetric loss. The resulting algorithms RIL and RIL_CO still require more optimal demonstrations than non-optimal ones in the dataset.

On the other hand, in many practical activities like sport games, it is common for people to use camera to record the excellent behaviors of athletes in terms of sequences of pictures and videos. However, it usually be hard to get the exact action label for each picture. To leverage such data for imitation learning, we need to recruit annotators to output action labels for pictures in the sequence. Limited by the quality of the annotators, action noise always occur during the action-labeling procedure. An amateur annotator may randomly pick an action for a picture that contains a state he never seen before. In this situation, the final demonstration will contain *state-independent action noise*. Besides, even an expert annotator makes mistakes. This is especially true when the annotator meets some similar and confusing states. In this situation, the annotator will output noisy actions that are dependent on the confusing states, resulting a demonstration with *state-dependent action noise*.

Previous methods (Tangkaratt et al., 2019; 2020; Zhang et al., 2021b) focus on imitation learning from a mixture of optimal and non-optimal demonstrations or noisy demonstrations with small noise

Figure 1: The full procedure of USN.

on the actions. They ignore the fact that both *state-independent action noise* and *state-independent action noise* widely exist in practice. Moreover, these methods are evaluated on low-dimensional environments, and are hard scale to high-dimensional environments. Therefore, existing methods usually fail in learning a good policy with action noise, especially in high-dimensional environments.

To tackle this challenge, we first conduct an investigation experiments to study the correlation between the dynamics of loss and predictive uncertainty metrics as the noise rate increase. Then, we propose a new method called uncertainty-aware sample-selection with soft negative learning (USN) based on the correlation. As shown in Figure 1, USN trains a policy with additional process of uncertainty-aware sample-selection for negative learning (USN). Specifically, in positive learning (marked in blue color), we train the IL model with any noise-tolerant loss function, and then estimate the predictive uncertainty measures during training. Then, we select large-loss samples using the estimated predictive uncertainty measure as the sampling threshold. This is motivated by recent works in the area of learning with noisy labels (Angluin & Laird, 1988; Smyth et al., 1994; Kalai & Servedio, 2005; Natarajan et al., 2013; Manwani & Sastry, 2013). Since deep networks learn easy patterns first (Arpit et al., 2017), they would first memorize training data of clean labels and then those of noisy labels with the assumption that clean labels are of the majority in a noisy class. Therefore, the large-loss samples can be regarded as noisy actions with high probability (Han et al., 2018; Yu et al., 2019; Wei et al., 2020; Yao et al., 2020) . Next, we employ negative learning (marked in green color) (Kim et al., 2019) on the automatically selected large-loss samples, along with positive learning on the full dataset.

USN is a meta-algorithm that is scalable to both offline imitation learning an online imitation learning. Compared with existing IL methods, our method has advantages in many aspects, including imitation performance, adaptivity and scalability . Our method can adaptively select large-loss samples for soft negative learning across different noise rates. Additionally, our method does not depend on any extra datasets, models or prior information about the noise model. That means our method avoids the extra efforts and drawbacks of estimating noise rates and transition matrix as previous noise-robust methods. Empirical results show that USN is scalable to Behavioral Cloning, online imitation learning and offline imitation learning.

## 2 BACKGROUND AND RELATED WORK

In this section, we firstly discuss existing offline and online imitation learning methods. Then, we discussed related works of learning with noisy labels.

**Behavioral cloning (BC)** is probably the simplest offline imitation learning algorithm. For imitation learning in environments with discrete action space, the BC policy $\pi(a|s)$ is optimized by softmax cross-entropy loss.

### 2.1 ONLINE IMITATION LEARNING

**Generative adversarial imitation learning (GAIL)** (Ho & Ermon, 2016) is one of the state-of-the-art online IL methods. GAIL treats imitation learning as a distribution matching problem. Built of top of GAN (Goodfellow et al., 2014), GAIL and its many (robust) variants (Li et al., 2017; Peng et al., 2019; Tangkaratt et al., 2019; 2020; Wang et al., 2021) has achieved great success in imitation learning in low-dimensional space even with noisy demonstrations. However, GAIL fails to scale

to high-dimensional imitation learning tasks (Brown et al., 2019a; Tucker et al., 2018). One simple solution to alleviate this issue is to initiate the actor of GAIL using behavior cloning.

**Selective Adversarial Imitation Learning (SAIL)** (Wang et al., 2021) was proposed to address imperfect demonstration issue, in which good demonstrations can be adaptively selected for training while bad demonstrations are abandoned. Specifically, a binary weight $w \in \{0, 1\}$ is assigned to each expert demonstration to indicate whether to select it for training. The weight is set to be determined by the reward function in Wasserstein GAIL (Xiao et al., 2019) (i.e. higher reward results in higher weight). The resulting algorithm - SAIL-hard is defined as follows:

$$\min_\theta \max_\phi \mathbb{E}_{(s,a)\sim\rho_E}[w(s,a)r_\phi(s,a) - Kw(s,a)] - \mathbb{E}_{(s,a)\sim\rho_\phi}[r_\phi(s,a)] + \lambda\Psi(r_\phi). \tag{1}$$

Besides hard binary weighting, they also propose a soft weighting scheme with the suggested optimal soft weight as $w^* = \frac{1}{1+e^{K-r_\phi(s,a)}}$. The corresponding algorithm - SAIL-soft is defined as follows:

$$\min_\theta \max_\phi \mathbb{E}_{(s,a)\sim\rho_E}[w(s,a)r_\phi(s,a) + f(w(s,a))] - \mathbb{E}_{(s,a)\sim\rho_\phi}[r_\phi(s,a)] + \lambda\Psi(r_\phi) \tag{2}$$

where $f(w) = w\log(w^{-1}-1) - \log(1-w) - Kw$. We regards GAIL, and its variants RIL, RIL_CO (Tangkaratt et al., 2020) and SAIL as baselines, for comparing their robustness against action noise.

## 2.2 OFFLINE IMITATION LEARNING

**Batch-constrained Q-learning (BCQ)** (Fujimoto et al., 2019b) is the first continuous control algorithm capable of learning from arbitrary batch data, without exploration. BCQ aims to perform Q-learning while constraining the action space to eliminate actions that are unlikely to be selected by the behavioral policy $\pi_b$, and are therefore unlikely to be contained in the batch. At its core, BCQ uses a state-conditioned generative model $G : s \rightarrow a$ to model the distribution of data in the batch, $G \approx b$ akin to a behavioral cloning model. As it is easier to sample from $\pi_b(a|s)$ than model $\pi_b(a|s)$ exactly in a continuous action space, the policy is defined by sampling $N$ actions $a_i$ from $G(s)$ and selecting the highest valued action according to a Q-network.

Discrete BCQ (Fujimoto et al., 2019a) extends BCQ to discrete action spaces by computing the probabilities of every action $G \approx \pi_b(a|s)$, and instead utilizing some threshold to eliminate actions. To adaptively adjust this threshold, Fujimoto et al. (2019a) scale it by the maximum probability from the generative model over all actions, to only allow actions whose relative probability is above some threshold. Specifically, Fujimoto et al. (2019a) applies Double DQN (Van Hasselt et al., 2016) to select the max valued action with the current Q-network $Q_\theta$, and evaluate with the target Q-network $Q_{\theta'}$. This results in an algorithm comparable to DQN (Mnih et al., 2015) where the policy is defined by a constrained argmax. The Q-network is trained by swapping the max operation with actions selected by the policy:

$$\mathcal{L}(\theta) = l_\kappa\left(r + \gamma \max_{a|G(a|s)/\max \hat{a}G(\hat{a}|s)>\tau} Q_{\theta'}(s', a') - Q_\theta(s, a)\right), \tag{3}$$

where $l_\kappa$ defines the Huber loss (Watkins, 1989):

$$l_\kappa(\delta) = \begin{cases} 0.5\delta^2 & \text{if } \delta \leq \kappa \\ \kappa(|\delta| - 0.5\kappa) & \text{otherwise.} \end{cases} \tag{4}$$

With this threshold $\tau$, Fujimoto et al. (2019a) maintain the original property of BCQ where setting $\tau = 0$ returns Q-learning and $\tau = 1$ returns an imitator of the actions contained in the batch. The generative model $G$, effectively a behavioral cloning network, is trained in a standard supervised learning fashion, with a cross-entropy loss. In this paper, we adopt the (Discrete) BCQ method for offline IL by training a intrinsic curiosity module (ICM) Pathak et al. (2017) for generating intrinsic rewards, instead of using the true reward data in the demonstration.

**Uncertainty weighted actor-critic (UWAC)** (Wu et al., 2021) is a recently proposed offline reinforcement learning algorithm that is able to detects OOD state-action pairs and down-weights their contribution in the training objectives accordingly. UWAC is closely related to our work, since it also uses the uncertainty estimation to improve the model robustness. Specifically, UWAC uses Monte Carlo (MC) dropout (Gal & Ghahramani, 2016) to estimate uncertainty of $Q$ model. The model

uncertainty is captured by the approximate predictive variance with respect to the estimated $\hat{Q}$ for $T$ stochastic forward passes:

$$Var[Q(s,a)] \approx \sigma^2 + \frac{1}{T}\sum_{t=1}^{T}\hat{Q}_t(s,a)^\top\hat{Q}_t(s,a) - \mathbb{E}[\hat{Q}(s,a)]^\top\mathbb{E}[\hat{Q}(s,a)]. \tag{5}$$

Then, UWAC down-weighs the Bellman loss for the Q function by inverse the uncertainty of the Q-target $Q_{\theta'}(s',a')$:

$$\mathcal{L}(Q_\theta) = \mathbb{E}_{(s'|s,a)\sim\mathcal{D}}\mathbb{E}_{a'\sim\pi(\cdot|s')}\Big[\frac{\beta}{Var[Q_{\theta'}(s',a')]}Err(s,a,s',a')^2\Big], \tag{6}$$

where $Err(s,a,s',a') = Q_\theta(s,a) - (R(s,a)) + \gamma Q_{\theta'}(s',a')$. This directly reduces the effect that OOD backups has on the overall training process (Wu et al., 2021). We treat UWAC as a strong baseline in our paper, and we also show that our USN can scale to MC dropout for uncertainty estimation. The original UWAC was built on top of BEAR (Kumar et al., 2019). To enable a fair comparison, we implement UWAC on top of BCQ as follows:

$$\mathcal{L}(\theta) = l_\kappa\left(r + \gamma\max_{a|G(a|s)/\max\hat{a}G(\hat{a}|s)>\tau}\frac{\beta}{Var[Q_{\theta'}(s',a')]}\big[Q_{\theta'}(s',a') - Q_\theta(s,a)\big]\right). \tag{7}$$

## 2.3 Learning with noisy labels

Learning with noisy labels aims to learn a robust classifier $f$ by exploiting training samples with only noisy labels $(x_i, \tilde{y}_i)_{i=1}^N$.

**Generalized Cross Entropy (GCE)** (Zhang & Sabuncu, 2018) is a generalization of Cross Entropy (CE) and Mean Absolute Error (MAE). To exploit the benefits of both the noise-robustness provided by MAE and the implicit weighting scheme of CCE, they proposed using the negative Box-Cox transformation (Box & Cox, 1964) as a loss function:

$$\mathcal{L}_q(f(x), e_j) = \frac{(1 - f_j(x)^q)}{q}, \tag{8}$$

where $q \in (0, 1]$. It is equivalent to CE for $\lim_{q\to0}\mathcal{L}_q(f(x), e_j)$, and becomes MAE/unhinged loss when $q = 1$.

**Label smoothing** has been commonly used to improve the performance of deep learning models (Szegedy et al., 2016; Chorowski & Jaitly, 2016; Vaswani et al., 2017; Zoph et al., 2018; Real et al., 2019; Huang et al., 2019). It serves as a regularizer (Li et al., 2020; Pereyra et al., 2017; Müller et al., 2019; Yuan et al., 2020; Zhang et al., 2021a) that improves the generalization and model calibration (predictive uncertainty) by using smoothed labels $\tilde{y}$ that mix the original one-hot labels $y$ with an uniform mixture over all possible labels for $\alpha \in [0, 1]$:

$$\tilde{y}_i = (1 - \alpha) \cdot y_i + \frac{\alpha}{K} \cdot \mathbf{1}, \tag{9}$$

where $K$ is the number of classes. Recently, Lukasik et al. (2020) demonstrated label smoothing is also effective when learning with symmetric label noise.

## 3 How does action noise affect imitation learning?

In this section, we first introduce *state-independent action noise* and *state-dependent action noise*. Then, we briefly summarize loss estimation and uncertainty estimation methods. Next, we provide a comprehensive study on the correlations between loss and uncertainty under multiple action noise.

### 3.1 Action noise

**State-independent action noise:** In traditional imitation learning, we consider the demonstrations $\mathcal{D} = \{s_i, a_i, s_{i+1}\}_{i=1}^N$ are collected from some expert demonstrator. In practice, we may recruit annotators to give action labels for the recorded sequences of expert behaviors. The *state-independent*

*action noise* occurs when a amateur annotator randomly pick an action for unseen states. The noise generation process is quite similar to the class-conditional label noise, where a corrupted label is randomly flipping from other classes (Symmetric noise) or its neighbor (Pairflip noise) (Natarajan et al., 2013; Han et al., 2018). Therefore, we build the noise modeling of *state-independent action noise* by borrowing the setting of class-conditional label noise. The resulting noisy demonstrations is noted as $\tilde{\mathcal{D}} = \{s_i, a_i, s_{i+1}\}_{i=1}^{N}$.

**State-dependent action noise:** The *state-dependent action noise* occurs when an expert annotator meets confusing states. In this situation, we assume the noisy action is dependent on the state feature and is independent on the expert action: $P(\tilde{a}_i|a_i, s_i) = P(\tilde{a}_i|s_i)$. Figure 2 shows a graphical model for *state-dependent action noise*. We generate the *state-dependent action noise* following the instance-dependent label noise setting in the Appendix B.

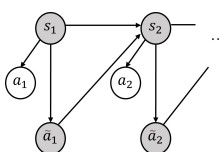

Figure 2: *State-dependent action noise*.

### 3.2 Correlation between loss and predictive uncertainty under action noise

In this paper, we provide a comprehensive study about the correlation between the loss, uncertainty estimation of Behavioral Cloning, online imitation learning (GAIL) and offline imitation learning (BCQ) models under diverse action noise in the demonstrations. We consider two types of noisy demonstrations: the first type of demonstration is the demonstrations with *state-independent* action noise; and the second type of demonstrations with *state-dependent* action noise. To study the correlations, we pre-train an imitation learning model with the optimal demonstration, and then estimate the corresponding loss and uncertainty on multiple noisy demonstrations.

**Loss estimation.** For the Behavioral Cloning model, GAIL actor and BCQ generative model, we use the cross-entropy as the loss estimation method. For the GAIL discriminator, we use the adversarial loss for the expert data for loss estimation.

**Uncertainty estimation.** Uncertainty estimation has been widely used in machine learning applications, as a complement that reflect the degree of trust in the model predictions (Kotelevskii et al., 2022). Usually, the total uncertainty of a prediction comes from two types of uncertainty: *aleatoric* and *epistemic* (Der Kiureghian & Ditlevsen, 2009; Kendall & Gal, 2017). The *aleatoric* uncertainty, known as data uncertainty, reflects the noise and ambiguity in the data. The *epistemic* uncertainty, known as model uncertainty, is related to the lack of knowledge about model parameters. Quantifying both types of uncertainty is crucial for safe decision in practical applications (Filos et al., 2020).

One simple *aleatoric* uncertainty is called MaxProb, which uses maximum softmax probabilities of deep neural network as the uncertainty measure (Nguyen et al., 2015). MC_dropout is based on Bayesian techniques, capturing both types of uncertainty and is known to be a more reliable uncertainty estimation method (Gal & Ghahramani, 2016). Recently, a promising direction of uncertainty estimation methods based on a single deterministic neural network has been developed (Lee et al., 2018; Liu et al., 2020). More recently, Kotelevskii et al. (2022) proposes a nonparametric uncertainty quantification (NUQ) method, which enables uncertainty to be disentangled into *aleatoric* and *epistemic* uncertainty, and is scalable to large dataset.

Expected Calibration Error (ECE) (Naeini et al., 2015; Guo et al., 2017) is also widely used to measure the predictive uncertainty (model calibration) of a deep network. To approximate the calibration error in expectation, ECE discretizes the probability interval into a fixed number of bins, and assigns each predicted probability to the bin that encompasses it. The calibration error is the difference between the fraction of predictions in the bin that are correct (accuracy) and the mean of the probabilities in the bin (confidence). More variants of calibration errors are summarized in the Appendix G.

Instead of using the uncertainty for OoD detection (Malinin & Gales, 2018), or down-weighting the Bellman loss for Q function (Wu et al., 2021) in RL, we leverage uncertainty for robust imitation learning in a novel direction. Namely, we first study the correlation between loss and uncertainty; and then propose an robust imitation learning paradigm called uncertainty-aware sample selection with negative learning (USN) based on the correlations. Our method USN is scalable to Behavioral Cloning, online imitation learning and offline imitation learning.

### 3.2.1 A COMPREHENSIVE STUDY ON THE CORRELATIONS OF LOSS AND UNCERTAINTY

Due to the space limit, we only provide samples of the correlations for BC, online IL and offline IL in this section. For more comprehensive results, please check the Appendix C.

**Behavioral Cloning.** Figure 3 shows that the loss has positive correlation with the ECE uncertainty as the noise rate increases. This results indicate that the loss and ECE uncertainty of BC model are good identifier of noisy actions.

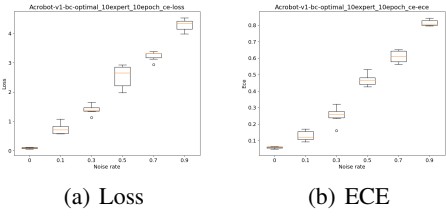

(a) Loss        (b) ECE

Figure 3: The correlations between loss, uncertainty of BC and noise rate of the noisy demonstration in the Acrobot-v1 task.

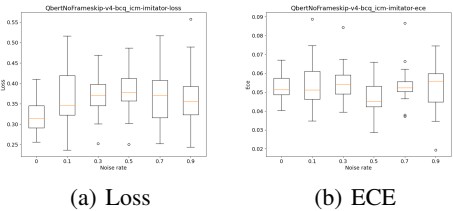

(a) Loss        (b) ECE

Figure 4: The correlations between loss, uncertainty of BCQ generative model and noise rate of the noisy demonstration in the Q*bert game.

**Online Imitation Learning.** Figure 5 and Figure 6 show that the loss has positive correlation with the ECE uncertainty as the noise rate increases. This results indicate that the loss and ECE uncertainty of GAIL actor and discriminator are good identifier of noisy actions.

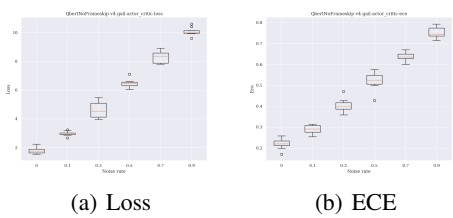

(a) Loss        (b) ECE

Figure 5: The correlations between loss, uncertainty of GAIL actor and noise rate of the noisy demonstration in the Q*bert game.

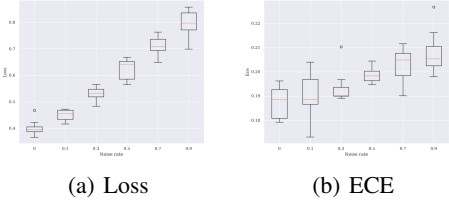

(a) Loss        (b) ECE

Figure 6: The correlations between loss, uncertainty of GAIL discriminator and noise rate of the noisy demonstration in the Q*bert game.

**Offline Imitation Learning.** We train the BCQ (Algorithm 2, Appendix A) with an ICM module (Pathak et al., 2017) for generating intrinsic rewards as the base offline imitation learning method. BCQ starts by sampling a mini batch of transitions from the input demonstration data $\mathcal{D}$. Then, we use the generative model $G$ to eliminate actions (step 5) for updating $Q$ network (step 6). The generative model $G$ is trained using a cross-entropy loss in step 7.

Figure 4 shows that the loss of the generative model $G$ goes up first and then drops as the noise rate increases. Interestingly, the ECE uncertainty drops at first and then increase as the noise rate increases, which shows a negative correlation with the dynamics of $G$'s loss.

**Analysis.** From Figure 3, Figure 4, Figure 5, Figure 6 and more results in the Appendix C, we can conclude that the loss estimation can be used as a criteria for detecting noisy actions using the BC model, BCQ's generative model, GAIL actor and discriminator. Specifically, the 'large-loss' samples have high probability to contain noisy actions. For the BC model, GAIL actor and GAIL discriminator, the uncertainty estimation ECE can also be used as a good criterion for detecting noisy actions. However for the offline imitation learning method BCQ, ECE does not show a clear positive correlation with the increase of action noise. Instead, the dynamics of ECE in BCQ's generative model has a negative correlation to the loss. In this paper, we focus on proposing a general method that detects noisy actions with high probability, and leverage the selected data to improve the robustness of BC, online IL and offline IL against diverse action noise in the demonstrations. To this end, we propose a general paradigm called Uncertainty-aware Sample-selection with Negative learning (USN), for robust training IL models against action noise. USN selects large-loss samples using the uncertainty estimation, e.g. ECE as a threshold that decides the number of sample selection.

## 4 METHOD

---

**Algorithm 1** Uncertainty-aware Sample-selection with Soft Negative learning (USN)

1: **Input:** A mini-batch $M$ of $B$ transitions $(s, a, s')$ from $\mathcal{D}$, negative learning loss weight as $\lambda_{neg}$, model parameter $\omega$.
2: Estimate predictive uncertainty, e.g. ECE, and regards it as the sample-selection threshold $\tau_u$.
3: Sample the large-loss batch $\tilde{M}$ by selecting the large-loss samples with a length of: $\#neg = (1 - \tau_u) * B$.
4: Generate complementary actions for $\tilde{M}$: $\bar{a} = $ Randomly select from $\{1, .., |\mathcal{A}|\} \backslash \{a\}$, resulting a complementary batch $\bar{M}$ for negative learning.
5: $\omega \leftarrow \arg\min_\omega \sum_{(s,a)\in M} \mathcal{L}_{\text{pos}}(s, a) + \lambda_{neg} \sum_{(s,a)\in \bar{M}} \mathcal{L}_{\text{neg}}(s, a)$.
6: **Output:** $w$.

---

Algorithm 1 summarizes the full procedure of USN. Our main goal is to develop a general robust IL paradigm against diverse type of action noise. To achieve this goal, we design USN to composite of two main steps: *uncertainty-aware sample-selection* and *negative learning for loss correction*.

**Uncertainty-aware sample-selection** aims to select samples that contains noisy actions with high probability. Given a mini-batch of demonstration data with a size of $B$, we first employ a basic IL method with parameter $\omega$, for positive learning on the full-batch data. Then in second step, we estimate the uncertainty, e.g. on the batch data and regards it as a threshold $\tau_u$ for selecting large-loss samples. Next, we sample the large-loss batch $\tilde{M}$ by selecting the large-loss samples with a length of: $\#neg = (1 - \tau_u) * B$. This procedure is called *uncertainty-aware sample-selection*.

Previous sample selection methods (Hafner et al., 2018; Han et al., 2020; Xia et al., 2021) usually assume that the noise rate is known, and design the sample selection threshold using the noise rate. In contrast, we define the sample selection threshold $\tau_u$ using the uncertainty estimation (e.g. ECE). Since the uncertainty estimation dynamically changes during training, the sample selection threshold is automatically updated and is adaptive to different noise rates. In this way, our method can maximally reduces the negative effects of action noise using the adaptive threshold, without requiring any prior knowledge about the noise model.

**Negative learning for loss correction.** Positive learning with full-batch data with noisy actions will results in bias in the loss training, and misguides the policy to choose the wrong actions. To correct the loss bias, we propose to leverage the selected large-loss sample for negative learning. Intuitively, the selected large-loss batch $\tilde{M}$ contains noisy actions with high probability. Its complementary set has more chances to contain true actions. Therefore, we generate complementary actions for $\tilde{M}$ by randomly selecting $\bar{a}$ from $\{1, .., |\mathcal{A}|\} \backslash \{a\}$, resulting a complementary bath $\bar{M}$ for negative learning. Negative learning on the complementary batch of the selected large-loss samples will correct the loss bias from action noise, and therefore improve the imitation learning performance. To further boost the performance, we implement negative learning with label smoothing, resulting *Soft Negative learning*. Specifically, we employ the following negative log likelihood (NLL) loss for negative learning on the 'large-loss' samples with label smoothing: $\mathcal{L}_{neg} = \text{NLL}\left(1 - G(a|s), (1 - \alpha) \cdot \bar{a} + \frac{\alpha}{|\mathcal{A}|} \cdot \mathbf{1}\right)$, where $\bar{a}$ is the complementary action of the 'large-loss' samples, and $\alpha$ is the smooth rate.

## 5 EXPERIMENTS AND RESULTS

In this section, we demonstrate how does USN improves the robustness of BC, onfline IL and offline IL when learning with *state-independent action noise* and *state-dependent action noise*. For Behavioral Cloning and BC initialized GAIL, we choose the BC as the basic IL model. For offline imitation learning, we obtain the basic IL model - BCQ-GCE, by applying GCE on the generative model: $\mathcal{L}_{pos} = \frac{(1 - G(a|s)^q)}{q}$. Algorithm 3 in the Appendix A presents an instance of our proposed robust IL method, BCQ-USN. BCQ-USN improves BCQ by updating the generative model $G$ using our USN method (Algorithm 1) in step 7. USN improves the ability of generative model $G$ for eliminating noisy actions for policy learning. Thus, BCQ-USN is more robust than BCQ and BCQ-GCE, achieving better imitation learning performance across diverse levels of action noise.

## 5.1 EXPERIMENTAL SETUP

We construct experiments on the widely used discrete control tasks and Atari games from OpenAI Gym environments. To evaluate the effectiveness of our methods, we pre-train expert agent and then generate demonstrations with state-dependent noisy actions using the pre-trained agent. For each setting, we have two demonstrations, $\mathcal{D}$ for *normal demonstration* and $\tilde{\mathcal{D}}$ for demonstration with state-dependent noisy actions only, a.k.a. the *noisy demonstration*. Note that we use ground-truth rewards only to train the expert agent, and we discard the rewards afterward.

In **setting 1**, we use the classic control tasks from OpenAI Gym - Acrobot-v1 and LunarLander-v2. We pre-train a DQN policy as the expert agent, and generate demonstrations with 50K steps. This setting is used to evaluate how our method USN improves Behavioral Cloning and offline imitation algorithms under *state-independent* action noise and *state-dependent* action noise. In **setting 2**, we use widely used Atari games. For online IL, we generate one full-episode demonstration using a PPO agent. For offline IL, we first pre-train a QRDQN policy as the expert agent. Then, we generate a 50K-step demonstration dataset. This setting is used for evaluation the robustness of our method USN on online imitation learning algorithm - GAIL, and offline imitation learning algorithms - BCQ and CQL against diverse action noise.

**Baseline algorithms:** For Behavioral Cloning, we compare USN with BC and GCE. For online imitation learning, we compare our method GAIL-USN to SOTA robust imitation learning algorithms: RIL_CO, SAIL amd the original GAIL. For offline imitation learning experiments, we compare BCQ-USN with BCQ in Algorithm 2 of Appendix A, which is trained with the rewards from an Intrinsic Curiosity Module (ICM). (b) BCQ-GCE, which is a robust BCQ by training generative model with GCE loss. and (c) UWAC, a SOTA robust offline imitation learning that also uses uncertainty estimation. We set the negative learning loss weight $\lambda_{neg} = 0.01$ for the classic control tasks and $\lambda_{neg} = 1.0$ for Atari games.

## 5.2 BEHAVIORAL CLONING WITH NOISY DEMONSTRATIONS

In the first experiment, we evaluate the robustness of our method USN on the classic control tasks - Acrobot-v1 and LunarLander-v2. Figure 7 and Figure 8 show the performance of BC, GCE and our USN under multiple *state-independent* action noise and *state-dependent* action noise, respectively. In both types of action noise in the demonstrations, our method USN achieves better performance than other baselines.

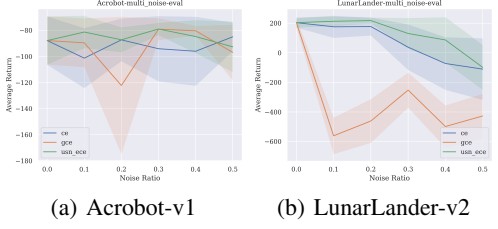
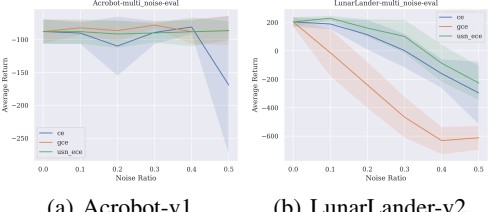

(a) Acrobot-v1    (b) LunarLander-v2       (a) Acrobot-v1    (b) LunarLander-v2

Figure 7: Average return of BC, GCE and our USN on classic control tasks with *state-independent* action noise.

Figure 8: Average return of BC, GCE and our USN on classic control tasks with *state-dependent* action noise.

## 5.3 ONLINE IMITATION LEARNING WITH NOISY DEMONSTRATIONS

Then, we show how our method USN can improves the robustness of online imitation learning against action noise. Since most of the previous robust online imitation learning algorithms (Tangkaratt et al., 2020; Wang et al., 2021) are based on GAIL, we also choose GAIL as our base method. However, the original GAIL can not achieve good performance in high-dimensional environment like Atari games (Brown et al., 2019a;b). Fortunately, we found that using Behavioral Cloning as an initialization for the actor, GAIL is able to achieve good performance on some Atari games, e.g. Seaquest, Q*bert and Hero with only one full-episode demonstration. We apply our method USN on the BC initialization and compares to SOTA robust imitation learning algorithms, RIL_CO and SAIL with soft weights.

The results under multiple *state-dependent* action noise for three Atari games are shown in Figure 9. SAIL totally fails in all the Atari games under action noise in the demonstrations. In the Seaquest game, both our method GAIL-USN and RIL_CO achieves the best performance; while RIL_CO fails in the other two games. Our method GAIL-USN outperforms all the baselines in the other two Atari games. More details are summarized in Appendix E.

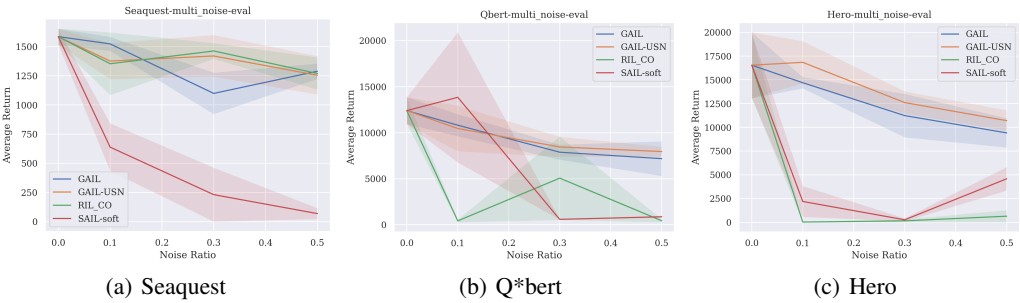

(a) Seaquest       (b) Q*bert       (c) Hero

Figure 9: The performance of GAIL and baselines with noisy demonstration across different *state-dependent* action noise on Atari games.

## 5.4 OFFLINE IMITATION LEARNING WITH NOISY DEMONSTRATIONS

Next, we evaluate the robustness of our method USN on offline imitation learning with demonstrations under both *state-independent* action noise and *state-dependent* action noise.

**Robustness to *state-independent action noise*.** In Figure 10(a)(b), we evaluate our method on two other types of **state-independent action noise**: symmetric noise and pairflip noise. These two types of noise have been widely studied in the literature of learning with noisy labels. We can see that the performance of BCQ drops when noise rate is larger than 0.45 in both Symmetric and Pairflip action noise. Our method BCQ-USN outperforms BCQ-GCE in both cases across diverse noise level. BCQ-USN-PC denotes our method using PC loss Ishida et al. (2017) as a negative learning loss. The results show that our method is scalable to multiple choices of negative learning loss.

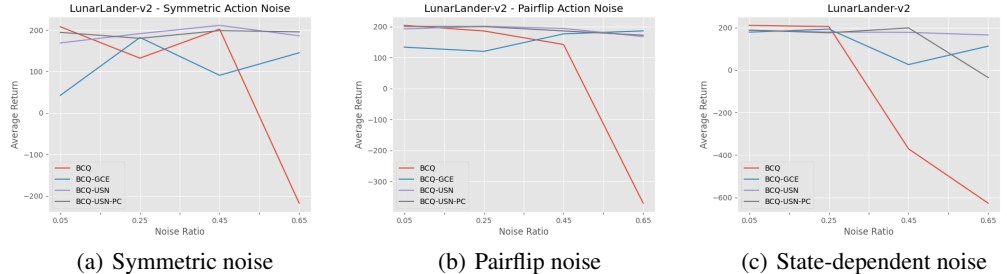

(a) Symmetric noise       (b) Pairflip noise       (c) State-dependent noise

Figure 10: Average return of BCQ, BCQ-GCE and our BCQ-USN on LunarLander-v2 with *state-independent* action noise (Symmetric and Pairflip), and *state-dependent* action noise.

**Robustness to *state-dependent action noise*.** Figure 10(c) shows the average return curves of BCQ, BCQ-GCE and our BCQ-USN on LunarLander-v2 across diverse levels of *state-dependent action noise*. For both types of action noise, BCQ-USN outperforms BCQ-GCE and BCQ across diverse noise levels. In Figure 34 of the Appendix, we show bar charts comparison of BCQ, BCQ-GCE and BCQ-USN for four Atari games. Our method, BCQ-USN consistently outperform BCQ-GCE and BCQ when the noise rate is 0.45. The performance improvements of BCQ-USN compared to other methods are usually significant. For this experiment, we compare our method with an related baseline - UWAC (Wu et al., 2021). In addition, we introduce a variants of our method called BCQ-USN-MC_dropout, which uses the MC dropout for uncertainty estimation in USN. The quantitative results are summarized in Table 1.

The results show that UWAC only outperforms BCQ-GCE on the Assault and AirRaid games, while performs much worse than BCQ-GCE on two other games. Our variant BCQ-USN-MC_dropout outperforms UWAC and BCQ-GCE on two games. This demonstrates that our method USN can

Table 1: Average return of BCQ, BCQ-GCE, BCQ-USN, BCQ-USN-MC_dropout, and UWAC when learning with *state-dependent action noise* with noise rate of 0.45 on four Atari games.

|  | Assault | Aline | AirRaid | NameThisGame |
|---|---|---|---|---|
| BCQ | 646.4 | 445.6 | 1,894.5 | 2,580.4 |
| BCQ-GCE | 667.4 | 655.8 | 1,858.5 | 2,615.2 |
| BCQ-USN (Ours) | 678.3 | **833.6** | 2,170.0 | **2,907.2** |
| BCQ-USN-MC_dropout (Ours) | **699.5** | 539.6 | **2,183.3** | 1,723.6 |
| UWAC (Wu et al., 2021) | 695.3 | 510.6 | **2,491.3** | 2,317.3 |

use MC dropout as an uncertainty estimator, while overall it performs worse than the variant using calibration errors. Overall, BCQ-USN outperform BCQ-GCE by 14% across the four Atari games.

Finally, we **generalize USN to Q-learning.** In the above experiments, USN is applied on top of Behavioral Cloning. To show the scalability of our method, we apply our USN method to a more general Q-learning based IL method without a generative model. We chose the conservative Q-learning (CQL) (Kumar et al., 2020) as our base method. In Appendix F, we implement a robust CQL algorithm, CQL-GCE, by training the Q network with the original conservative loss and an additional GCE loss. Similar to BCQ-USN, we apply USN on the positive learning baseline - CQL-GCE. Specifically, we estimate calibration error metrics for the Q network, and then use the calibration error to determine the large-loss samples for negative learning. Empirically, we evaluate the resulting algorithm CQL-USN on the Q*bert Atari game with *state-dependent action noise*, symmetric action noise and pairflip action noise across different noise rates. Figure 11 shows the comparative results of CQL, CQL-GCE and CQL-USN on the Q*bert game. Impressively, CQL-USN usually outperform CQL and CQL-GCE by a large margin. Therefore, our USN method has good scalability to general Q-learning based IL methods, without requiring the generative model.

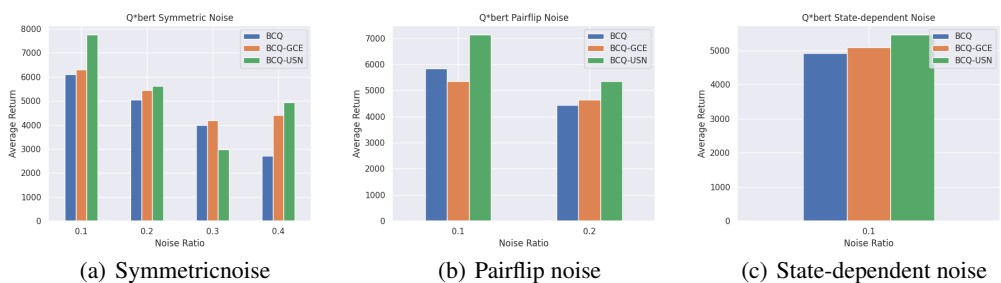

(a) Symmetricnoise  (b) Pairflip noise  (c) State-dependent noise

Figure 11: Average return of CQL, CQL-GCE and CQL-USN on Q*bert game with *state-indepedent* action noise (Symmetric and Pairflip) and *state-dependent* action noise.

## 6 CONCLUSION

In this paper, we have provided a comprehensive study of correlations between loss estimation and uncertainty estimation of imitation learning models. Then, based on the correlations, we propose an novel paradigm for robust imitation learning with action noise. Our method, uncertainty-aware sample selection for negative learning (USN), scales to Behavioral Cloning, online imitation learning and offline imitation learning with *state-independent action noise* and *state-dependent action noise*. Moreover, our USN paradigm can use diverse predictive uncertainty for robust imitation learning, showing the nice adapativity and scalability of our method. In addition, the ablation studies in the Appendix H also demonstrate that our method is not sensitive to the adaptation of the hyperparameters. In the near future, we are interested in extending our method to continuous control tasks by studying the correlations between loss and regression uncertainty estimations of IL models. Also, it is worthy to investigate the application of our method on actor-critic learning and real-world tasks.

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

## A    ALGORITHMS

Algorithm 2 and Algorithm 3 shows the procedure of BCQ and BCQ-USN for offline imitation learning. We implement our methods using the Tianshou platform (Weng et al., 2021) for offline imitation learning, and train all the algorithms with the default hyperparameters of BCQ.

---

**Algorithm 2** BCQ with ICM for offline imitation learning.

---

1: **Input:** Demonstration $\mathcal{D} = \{(s_i, a_i, s_{i+1})\}_{i=1}^{N}$, number of iterations $T$, target_update_rate, mini-batch size $B$, threshold $\tau$.
2: Initialize Q-network $Q_\theta$, generative model $G$ with parameter $\omega$ and target network $Q_{\theta'}$ with $\theta' \leftarrow \theta$.
3: **for** $t = 1$ to $T$ **do**
4:     Sample mini-batch $M$ of $B$ transitions $(s, a, s')$ from $\mathcal{D}$.
5:     Train an ICM module to generate intrinsic reward: $r = \text{ICM}(s, a, s')$.
6:     $a' \leftarrow \arg\max_{a' | G(a'|s') / \max \hat{a} G(\hat{a}|s') > \tau} Q_\theta(s', a')$.
7:     $\theta \leftarrow \arg\min_\theta \sum_{(s,a,s') \in M} k_\kappa(r + \gamma Q_{\theta'}(s', a') - Q_\theta(s, a))$
8:     $\omega \leftarrow \arg\min_\omega - \sum_{(s,a) \in M} \log G(a|s)$.          //Update $G$ using cross-entropy loss.
9:     If $t$ mod target_update_rate $= 0 : \theta' \leftarrow \theta$.
10: **end for**

---

---

**Algorithm 3** BCQ-USN for robust offline imitation learning against action noise.

---

1: **Input:** Demonstration $\mathcal{D} = \{(s_i, a_i, s_{i+1})\}_{i=1}^{N}$, number of iterations $T$, target_update_rate, mini-batch size $B$, threshold $\tau$.
2: Initialize Q-network $Q_\theta$, generative model $G$ with parameter $\omega$ and target network $Q_{\theta'}$ with $\theta' \leftarrow \theta$.
3: **for** $t = 1$ to $T$ **do**
4:     Sample mini-batch $M$ of $B$ transitions $(s, a, s')$ from $\mathcal{D}$.
5:     Train an ICM module to generate intrinsic reward: $r = \text{ICM}(s, a, s')$.
6:     $a' \leftarrow \arg\max_{a' | G(a'|s') / \max \hat{a} G(\hat{a}|s') > \tau} Q_\theta(s', a')$.
7:     $\theta \leftarrow \arg\min_\theta \sum_{(s,a,s') \in M} k_\kappa(r + \gamma Q_{\theta'}(s', a') - Q_\theta(s, a))$
8:     $\omega \leftarrow \text{USN}(M)$          // Update $G$ using USN approach in Algorithm 1.
9:     If $t$ mod target_update_rate $= 0 : \theta' \leftarrow \theta$.
10: **end for**

---

## B    STATE-DEPENDENT ACTION NOISE GENERATION

---

**Algorithm 4** State-Dependent Action Noise Generation

---

1: **Input:** Expert demonstration $\mathcal{D} = \{(s_i, a_i, s_{i+1})\}_{i=1}^{N}$; Noise rate: $\epsilon$; Size of feature: $1 \times |S|$; Number of classes: $|\mathcal{A}|$.
2: **Iterations:**
3: Sample instance flip rates $q_n$ from the truncated normal distribution $\mathcal{N}(\epsilon, 0.1^2, [0, 1])$;
4: Sample $W \in \mathcal{R}^{|S| \times |\mathcal{A}|}$ from the standard normal distribution $\mathcal{N}(0, 1^2)$;
5: **for** $n = 1$ to $N$ **do**
6:     $p = s_i \cdot W$          // Generate state dependent flip rates. The size of $p$ is $1 \times |\mathcal{A}|$.
7:     $p_{a_i} = -\infty$          // Only consider entries different from the expert actions
8:     $p = q_i \cdot \text{softmax}(p)$          // Let $q_n$ be the probability of getting a wrong action
9:     $p_{a_i} = 1 - q_i$          // Keep expert actions w.p. $1 - q_i$
10:     Randomly choose a action from the action space as noisy action $\tilde{a}_i$ according to $p$;
11: **end for**
12: **Output:** Noisy Demonstrations $\tilde{\mathcal{D}} = \{(s_i, \tilde{a}_i, s_{i+1})\}_{i=1}^{N}$.

---

We simulate the generation of *state-dependent* action noise by following the *instance-dependent* label noise setting (Xia et al., 2020). Note that it is more realistic that different states have different flip

rates. Without constraining different states to have a same flip rate, it is more challenging to model the label noise and train robust classifiers. In Step 1, in order to control the global flip rate as $\tau$ but without constraining all the states to have a same flip rate, we sample their flip rates from a truncated normal distribution $\mathcal{N}(\tau, 0.1^2, [0, 1])$. Specifically, this distribution limits the flip rates of states in the range [0, 1]. Their mean and standard deviation are equal to the mean $\tau$ and the standard deviation 0.1 of the selected truncated normal distribution respectively.

In Step 2, we sample parameters $w_1, w_2, \cdots, w_c$ from the standard normal distribution for generating *state-dependent action noise*. The dimensionality of each parameter is $d \times c$, where d denotes the dimensionality of the state. Learning these parameters is critical to model *state-dependent action noise*. However, it is hard to identify these parameters without any assumption. Note that an state with expert action $a$ will be flipped only according to the $a-$th row of the transition matrix. Thus, in Steps 4 to 7, we only use the $a_i-$th row of the state-dependent transition matrix for the state $s_i$. Specifically, Steps 5 and 7 are to ensure the diagonal entry of the $a_i-$th row is $1 - q_i$. Step 6 is to ensure that the sum of the off-diagonal entries is $q_i$.

## C  THE CORRELATION BETWEEN LOSS, UNCERTAINTY AND NOISE RATES OR OPTIMALITY OF DEMONSTRATIONS IN THE ATARI DOMAIN

### C.1  RESULTS OF BC ON CLASSIC CONTROL TASKS.

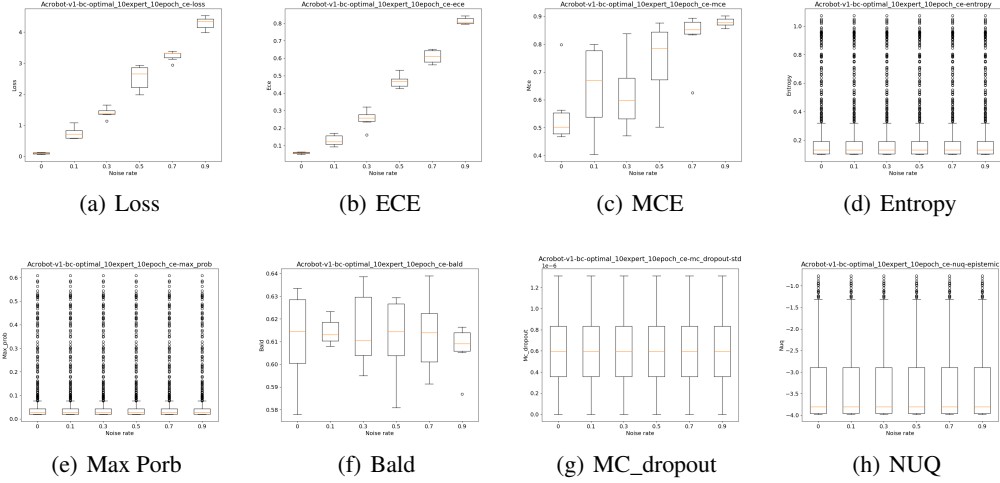

Figure 12: The correlations between loss, uncertainty of BC and noise rate of the noisy demonstration in the Acrobot-v1 task.

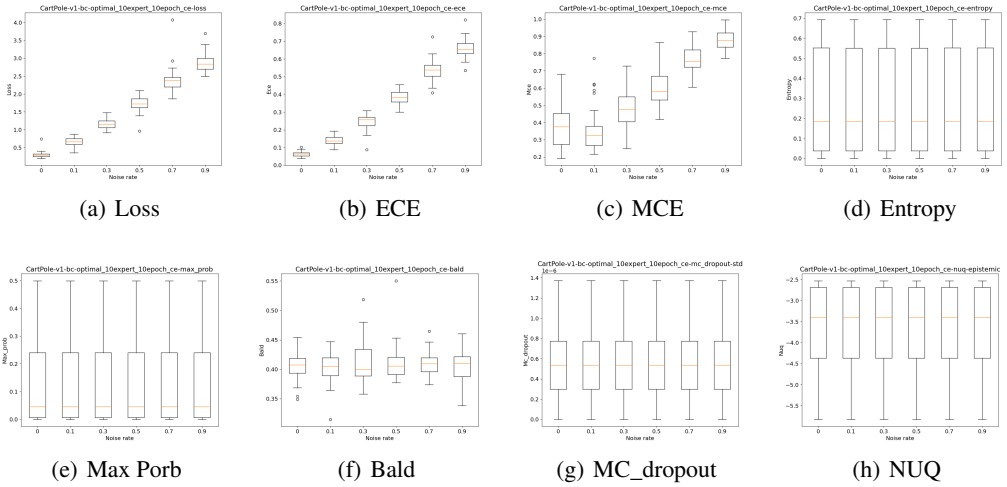

Figure 13: The correlations between loss, uncertainty of BC and noise rate of the noisy demonstration in the CartPole-v1 task.

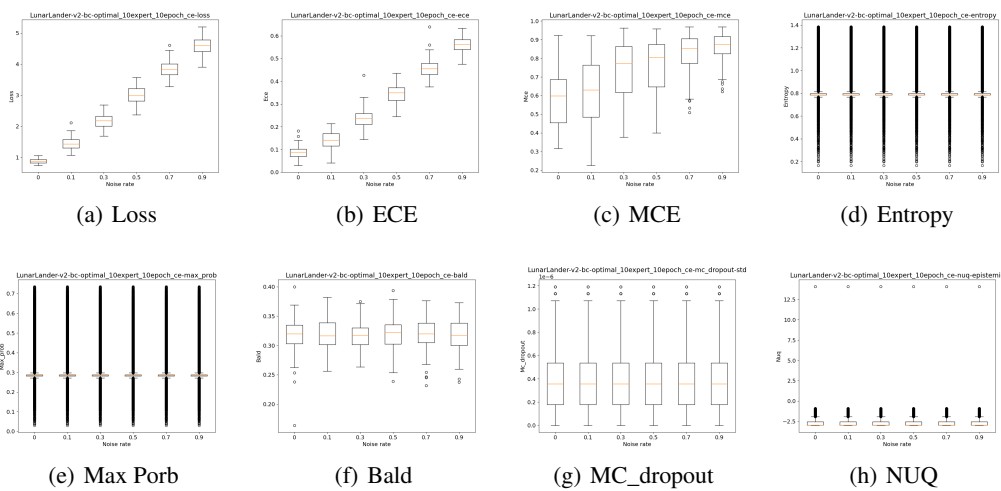

Figure 14: The correlations between loss, uncertainty of BC and noise rate of the noisy demonstration in the LunarLander-v2 task.

## C.2 Results of GAIL on Atari games

### C.2.1 Results of GAIL on a single Atari game.

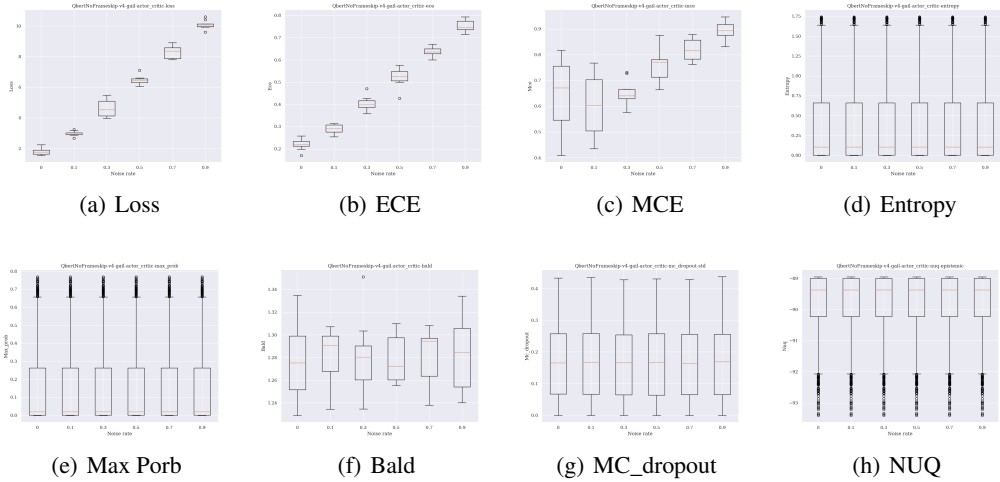

(a) Loss      (b) ECE      (c) MCE      (d) Entropy

(e) Max Porb      (f) Bald      (g) MC_dropout      (h) NUQ

Figure 15: The correlations between loss, uncertainty of GAIL actor and noise rate of the noisy demonstration in the Q*bert game.

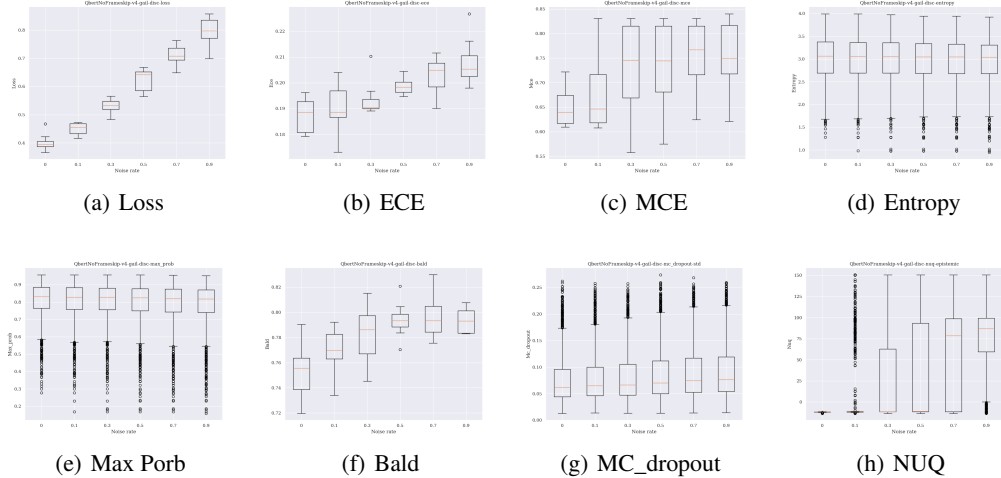

(a) Loss      (b) ECE      (c) MCE      (d) Entropy

(e) Max Porb      (f) Bald      (g) MC_dropout      (h) NUQ

Figure 16: The correlations between loss, uncertainty of GAIL discriminatorand noise rate of the noisy demonstration in the Q*bert game.

For the noisy demonstration with *state-dependent* action noise, we study correlation between loss, uncertainty estimation of GAIL and noise rate of the demonstrations. The results of GAIL actor and discriminator in Q*bert game are shown in Figure 15 and Figure 16. As expected, with the increasing of noise rates in the demonstrations, the loss estimation of the GAIL actor increases accordingly. For the uncertainty estimations, ECE and MCE show the positive correlation with the loss estimation and the increase of noise rate. For the GAIL discriminator, the loss estimation and three uncertainty estimations (i.e. ECE, MCE and NUQ) hold the positive correlation with the increase of noise rate in the demonstrations. The Bald and MC_dropout uncertainty estimations also show light positive correlations. The above observations hold for many Atari games. The results can be found bellow. Therefore, the loss estimation, the uncertainty estimations - ECE and MCE, can be used as criteria for detecting noisy actions in the demonstrations using the GAIL actor. Similarly, the loss estimation, the uncertainty estimations - ECE, MCE, NUQ and even Bald and MC_dropout, can be used as criteria for detecting noisy actions in the demonstrations using the GAIL discriminator.

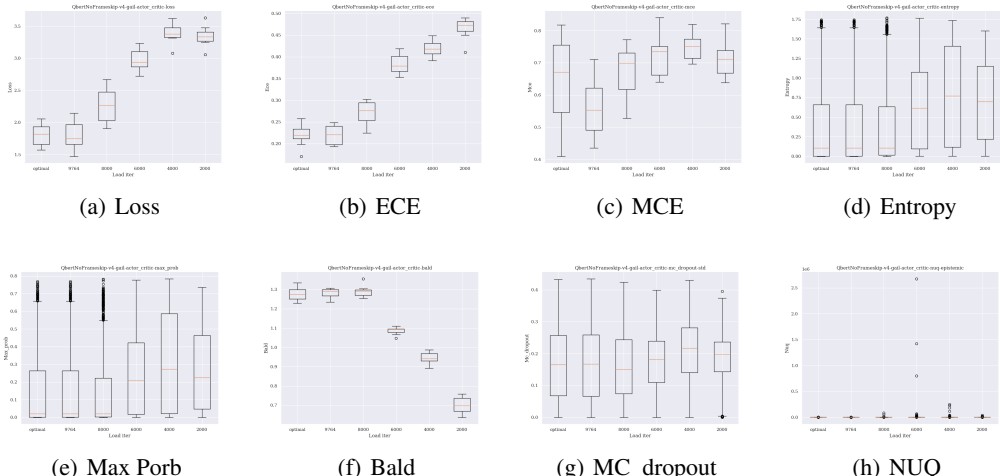

Figure 17: The correlations between loss, uncertainty of GAIL actor and optimality of the noisy demonstrations in the Q*bert game.

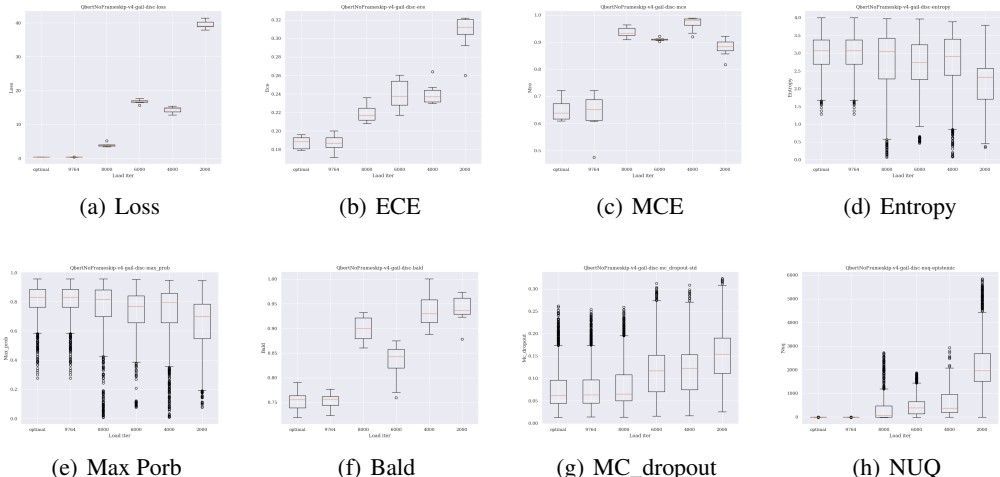

Figure 18: The correlations between loss, uncertainty of GAIL discriminatorand optimality of the noisy demonstrations in the Q*bert game.

For the noisy demonstration with suboptimal demonstrations, we study correlation between loss, uncertainty estimation of GAIL and optimality of the demonstrations. The results of GAIL actor and discriminator in Q*bert game are shown in Figure 17 and Figure 18. The lower load iter denotes worse optimality. The results show that with the decreasing of the optimality in the demonstrations, the loss estimation of the GAIL actor increases accordingly. For the uncertainty estimations, ECE and MCE show the positive correlation with the loss estimation and the increase of noise rate. Entropy, MaxProb and MC_dropout show light positive correlations. For the GAIL discriminator, the loss estimation and many uncertainty estimations (i.e. ECE, MCE, Bald, MC_dropout and NUQ) hold the positive correlation with the decrease of optimality in the demonstrations. The above observations hold for many Atari games. The results can be found bellow. Therefore, the loss estimation, the uncertainty estimations - ECE, MCE and even MC_dropout, can be used as creteria for detecting suboptimal data in the demonstrations using the GAIL actor. Similarly, the loss estimation, the uncertainty estimations - ECE, MCE, Bald, MC_dropout and NUQ, can be used as criteria for detecting suboptimal data in the demonstrations using the GAIL discriminator.

C.2.2 RESULTS OF GAIL ON MORE ATARI GAMES WITH NOISY ACTIONS

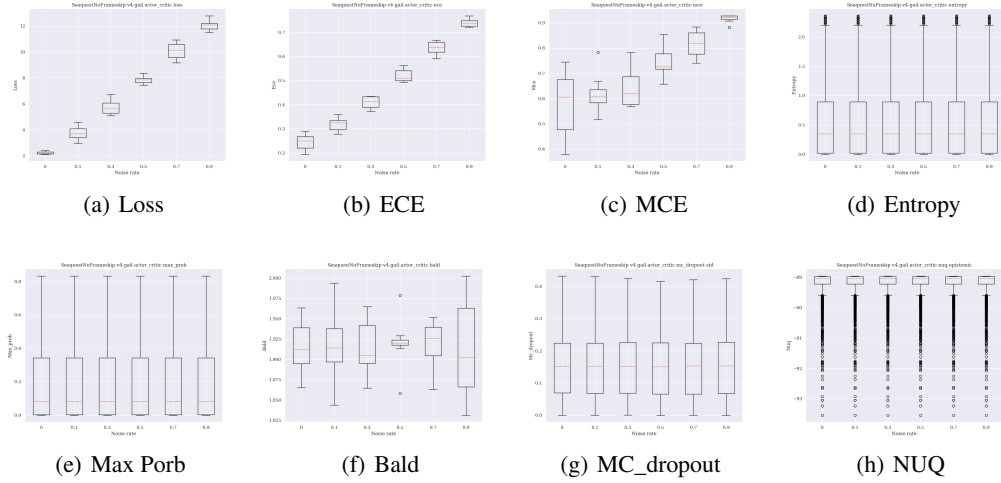

(a) Loss     (b) ECE     (c) MCE     (d) Entropy

(e) Max Porb     (f) Bald     (g) MC_dropout     (h) NUQ

Figure 19: The correlations between loss, uncertainty of GAIL actor and noise rate of the noisy demonstration in the Seaquest game.

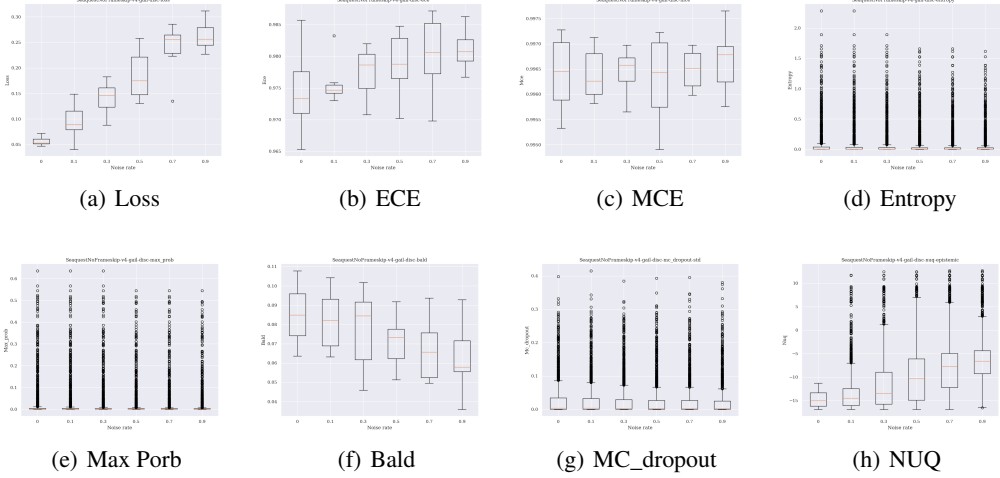

(a) Loss     (b) ECE     (c) MCE     (d) Entropy

(e) Max Porb     (f) Bald     (g) MC_dropout     (h) NUQ

Figure 20: The correlations between loss, uncertainty of GAIL discriminatorand noise rate of the noisy demonstration in the Seaquest game.

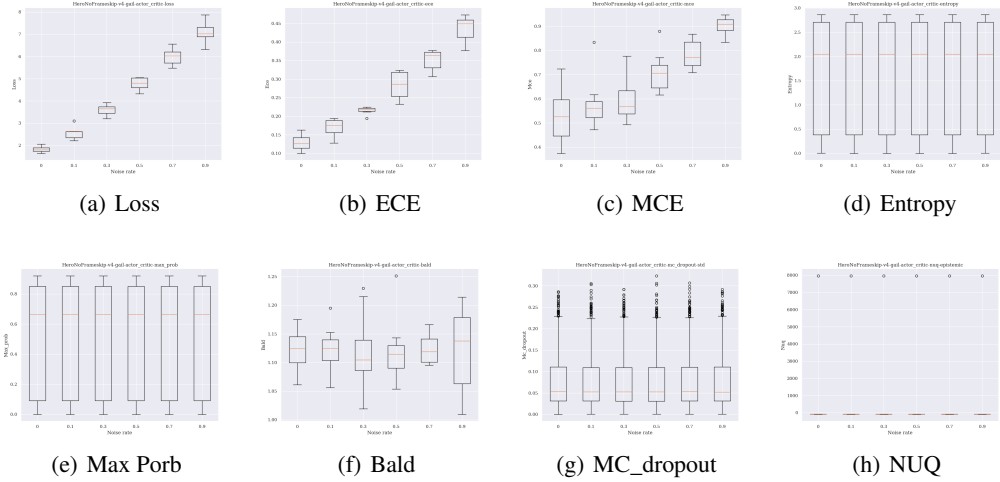

Figure 21: The correlations between loss, uncertainty of GAIL actor and noise rate of the noisy demonstration in the Hero game.

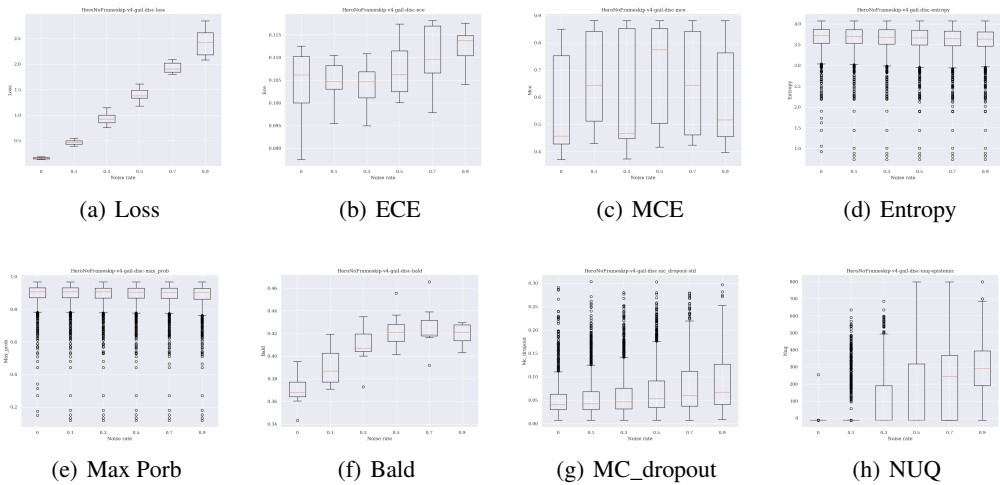

Figure 22: The correlations between loss, uncertainty of GAIL discriminatorand noise rate of the noisy demonstration in the Hero game.

### C.2.3 RESULTS OF GAIL ON MORE ATARI GAMES WITH SUB-OPTIMAL DEMONSTRATIONS

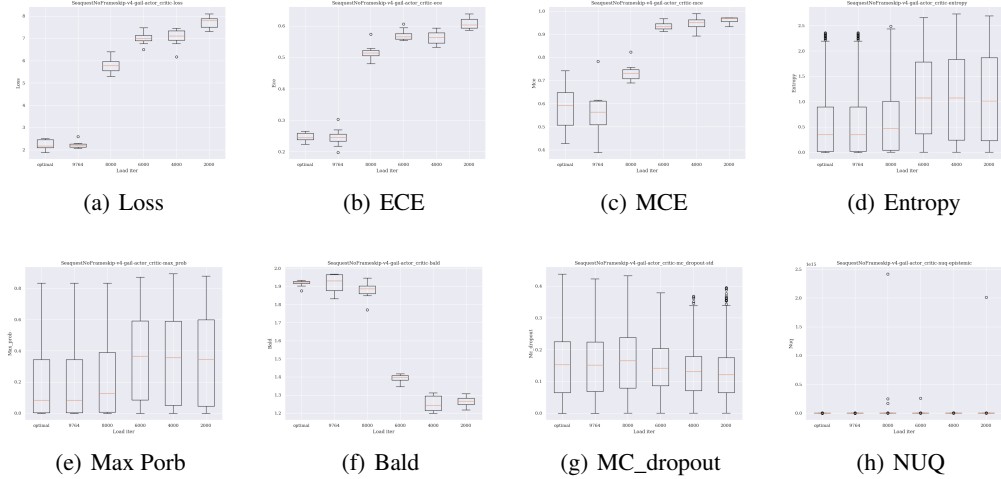

(a) Loss     (b) ECE     (c) MCE     (d) Entropy

(e) Max Porb     (f) Bald     (g) MC_dropout     (h) NUQ

Figure 23: The correlations between loss, uncertainty of GAIL actor and optimality of the noisy demonstrations in the Seaquest game.

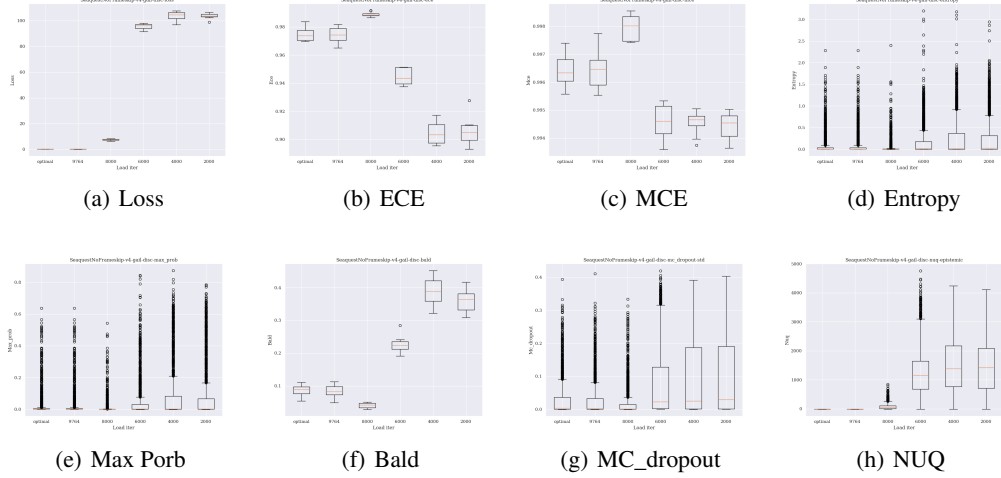

(a) Loss     (b) ECE     (c) MCE     (d) Entropy

(e) Max Porb     (f) Bald     (g) MC_dropout     (h) NUQ

Figure 24: The correlations between loss, uncertainty of GAIL discriminatorand optimality of the noisy demonstrations in the Seaquest game.

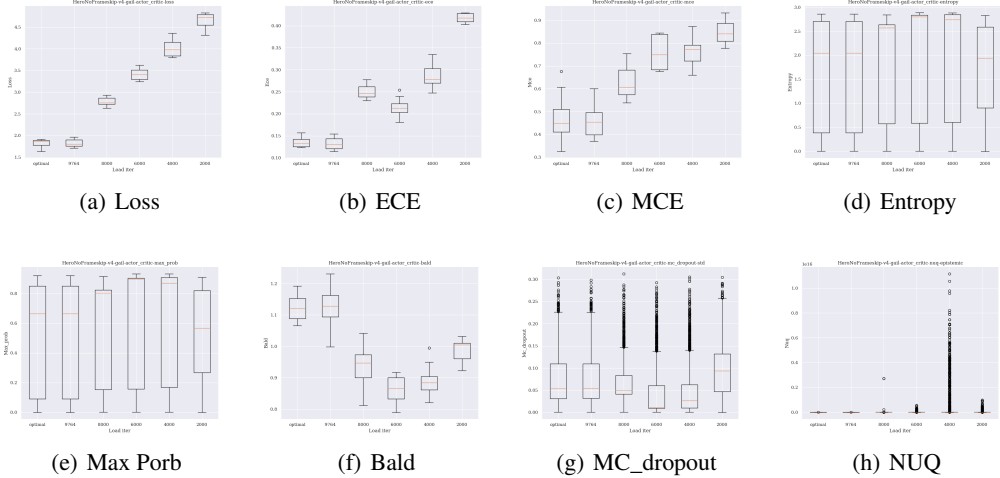

(a) Loss        (b) ECE        (c) MCE        (d) Entropy

(e) Max Porb        (f) Bald        (g) MC_dropout        (h) NUQ

Figure 25: The correlations between loss, uncertainty of GAIL actor and optimality of the noisy demonstrations in the Hero game.

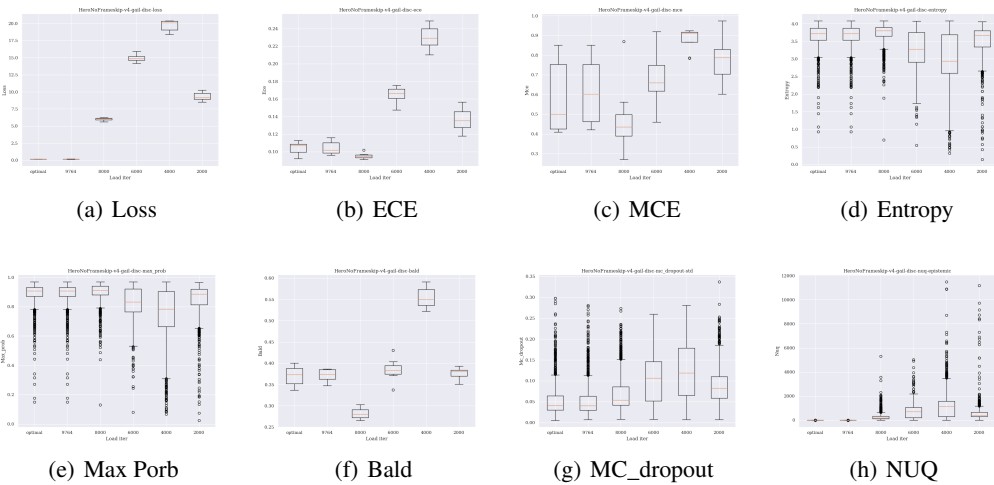

(a) Loss        (b) ECE        (c) MCE        (d) Entropy

(e) Max Porb        (f) Bald        (g) MC_dropout        (h) NUQ

Figure 26: The correlations between loss, uncertainty of GAIL discriminatorand optimality of the noisy demonstrations in the Hero game.

C.3 REUSULTS OF BCQ_ICM ON ATARI GAMES.

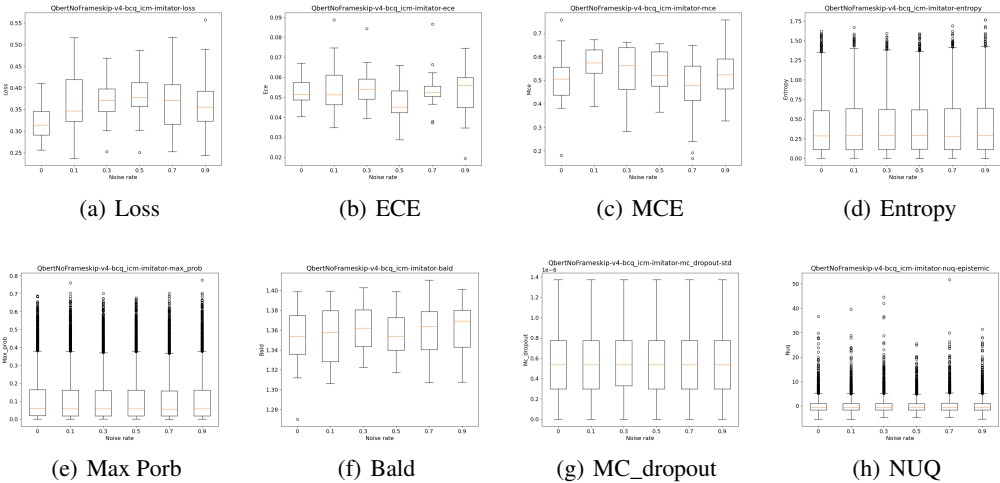

(a) Loss        (b) ECE        (c) MCE        (d) Entropy

(e) Max Porb        (f) Bald        (g) MC_dropout        (h) NUQ

Figure 27: The correlations between loss, uncertainty of BCQ_ICM imitator and noise rate of the noisy demonstration in the Q*bert game.

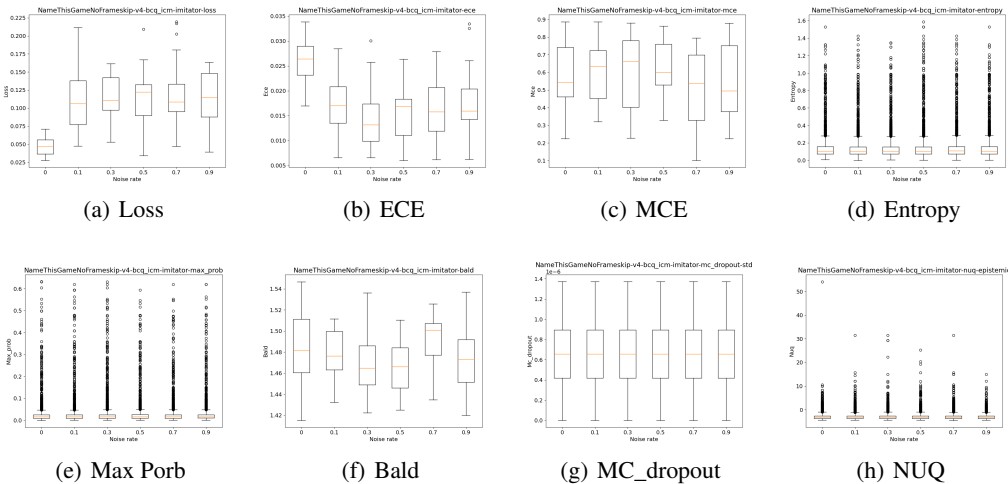

(a) Loss        (b) ECE        (c) MCE        (d) Entropy

(e) Max Porb        (f) Bald        (g) MC_dropout        (h) NUQ

Figure 28: The correlations between loss, uncertainty of BCQ_ICM imitator and noise rate of the noisy demonstration in the NameThisgame game.

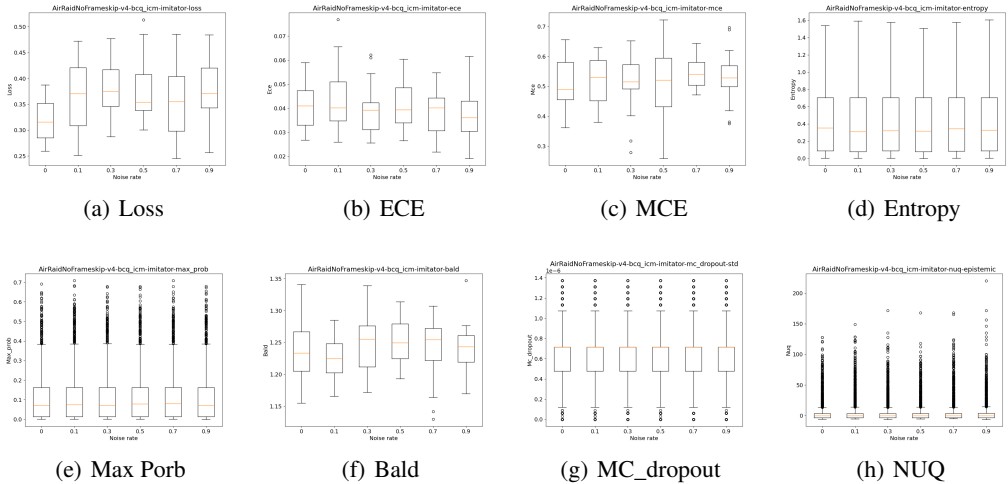

(a) Loss     (b) ECE     (c) MCE     (d) Entropy

(e) Max Porb     (f) Bald     (g) MC_dropout     (h) NUQ

Figure 29: The correlations between loss, uncertainty of BCQ_ICM imitator and noise rate of the noisy demonstration in the AirRaid game.

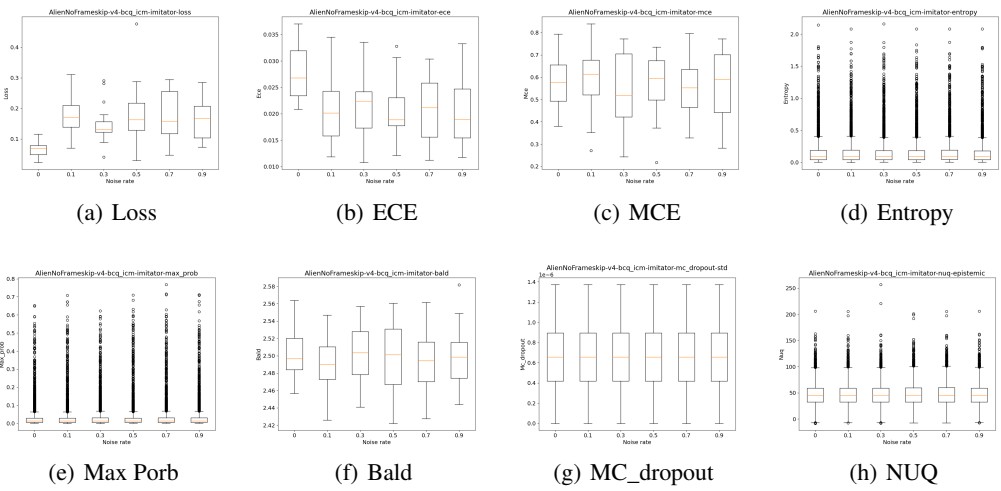

(a) Loss     (b) ECE     (c) MCE     (d) Entropy

(e) Max Porb     (f) Bald     (g) MC_dropout     (h) NUQ

Figure 30: The correlations between loss, uncertainty of BCQ_ICM imitator and noise rate of the noisy demonstration in the Alien game.

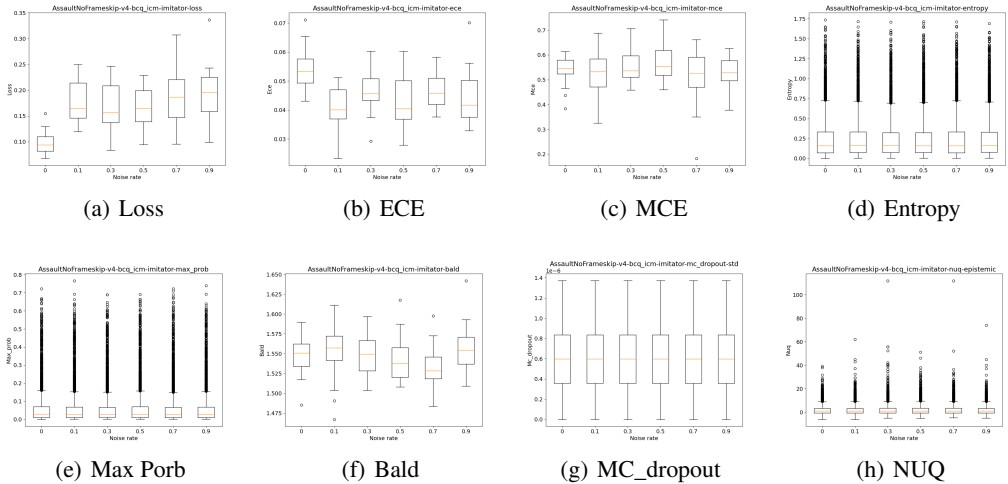

(a) Loss       (b) ECE       (c) MCE       (d) Entropy

(e) Max Porb       (f) Bald       (g) MC_dropout       (h) NUQ

Figure 31: The correlations between loss, uncertainty of BCQ_ICM imitator and noise rate of the noisy demonstration in the Assault game.

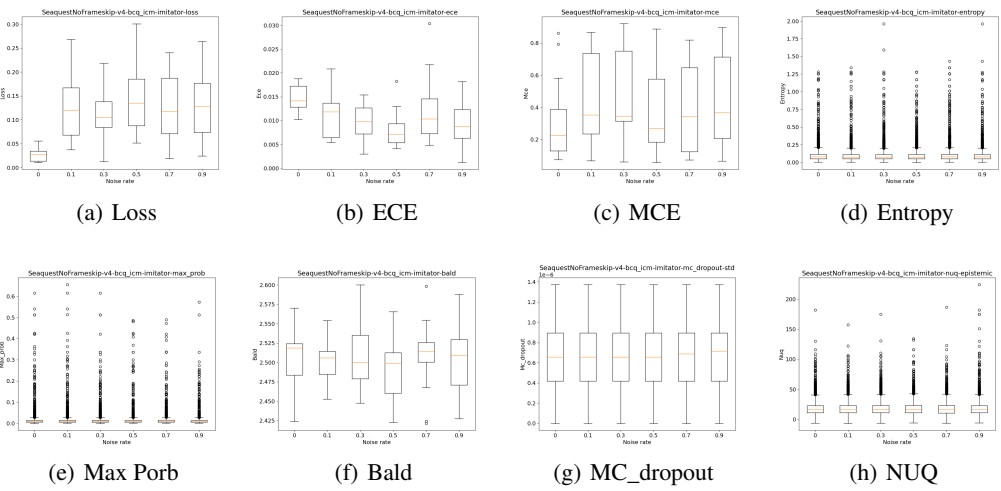

(a) Loss       (b) ECE       (c) MCE       (d) Entropy

(e) Max Porb       (f) Bald       (g) MC_dropout       (h) NUQ

Figure 32: The correlations between loss, uncertainty of BCQ_ICM imitator and noise rate of the noisy demonstration in the Seaquest game.

# D    ADDITIONAL RESULTS.

## D.1    COMPARATIVE RESULTS ACROSS MULTIPLE NOISE RATES.

Table 2 shows the comparative results of BCQ, BCQ-GCE and our BCQ-USN on LunarLander-v2 across different levels of *state-dependent action noise*.

Table 2: Average return of offline imitation learning on the LunarLander-v2 with diverse levels of *state-dependent action noise*.

| Method | Noise Rates | | | |
|---|---|---|---|---|
| | 0.05 | 0.25 | 0.45 | 0.65 |
| BCQ | **210.8** | 205.1 | -370.9 | -626.8 |
| BCQ-GCE | 178.7 | 193.4 | 26.1 | 113.0 |
| BCQ-USN-ECE | 177.2 | 193.7 | 0.9 | **167.1** |
| BCQ-USN-ACE | 148.4 | **211.1** | 160.4 | 50.8 |
| BCQ-USN-Max_CE | 188.0 | 179.2 | **177.9** | 165.5 |

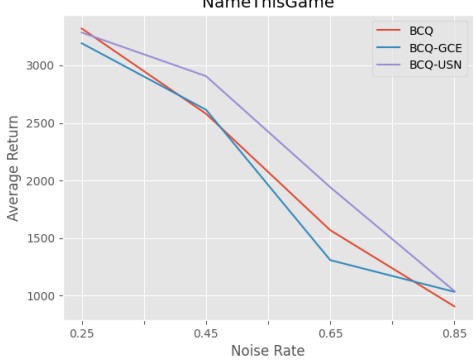

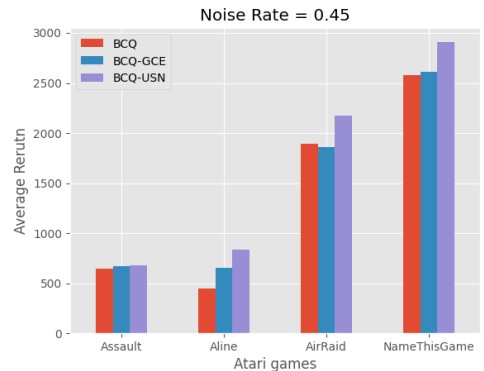

Figure 33: Average return of BCQ, BCQ-GCE and our BCQ-USN on LunarLander-v2 and NameThis-Game across diverse noise rates.

Figure 34: Bar chart comparison of average return of BCQ, BCQ-GCE and BCQ-USN in four Atari games with 45% *state-dependent action noise*.

Table 3: Average return of offline imitation learning on the Atari game - NameThisGame with diverse levels of *state-dependent action noise*.

| Game | Noise Rates | | | | |
|---|---|---|---|---|---|
| | 0.05 | 0.25 | 0.45 | 0.65 | 0.85 |
| BCQ | **3,640.0** | **3,320.4** | 2,580.4 | 1,569.0 | 906.6 |
| BCQ-GCE | 3,605.0 | 3,195.2 | 2,615.2 | 1,309.0 | 1,033.4 |
| BCQ-USN | 3,332.6 | 3,285.6 | **2,907.2** | **1,942.4** | **1,039.4** |

# E    ONLINE IMITATION LEARNING

## E.1    GENERATIVE ADVERSARIAL IMITATION LEARNING (GAIL)

Generative adversarial imitation learning (GAIL) was proposed based on the general framework of Generative Adversarial Networks (GANs) Goodfellow et al. (2014). GAIL regards the classical imitation learning problem as an occupancy measure matching between the expert policy and the agent policy. Specifically, GAIL uses a discriminator $D_\phi$ to distinguish expert transitions from agent

transitions, while the agent is to "fool" the discriminator into treating agent transitions as expert data. Formally, the objective function of GAIL is written as

$$\min_{\theta} \max_{\phi} \mathbb{E}_{(s,a)\sim\rho_{\theta}}[\log D_{\phi}(s,a)] + \mathbb{E}_{(s,a)\sim\rho_E}[\log(1 - D_{\phi}(s,a))] \tag{10}$$

where $\rho_{\theta}$ and $\rho_E$ denote the occupancy measures of agent policies $\pi_{\theta}$ and the expert $\pi_E$, respectively.

GAIL and its many variants (Peng et al., 2019; Li et al., 2017) has achieved great success in imitation learning in low-dimensional tasks. In imitation learning with imperfect demonstrations / noisy demonstrations, many previous methods are also based the framework of GAIL. However, GAIL had been demonstrated that does not scale well to high-dimensional tasks like Atari games (Tucker et al., 2018; Brown et al., 2019a;b). To boost the performance of GAIL in high-dimensional task, we take a simple step by initializing the actor with behavior cloning. We found that GAIL is able to achieve demonstration-lelve performance on some games by using the BC initialization on the actor. Our implementation is based on the 'gail_atari' repository [1].

### E.2 GAIL-USN

To extend our USN paradigm to GAIL, we directly apply USN on the behavior cloning model in the initialization step.

### E.3 EXPERIMENTS AND RESULTS

**Experiment setup.** For online imitation learning, we consider learning from the noisy environments. Namely, besides to the noisy demonstration data, the data collected from the environment are full of action noise. This happens in practice when the environments suffers from system error or the data collecting and transferring protocol under attack. This task is more challenging than previous imitation learning problem with noisy demonstrations only.

**Baselines.** In this experiment, we compare our method GAIL-USN to GAIL and the following state-of-the-art robust imitation learning methods:

- **RIL** and **RIL_CO** (Tangkaratt et al., 2020)
- **SAIL-soft** and **SAIL-hard** (Wang et al., 2021)

Since all the baselines are variants of GAIL, we use the same settings including behavior cloning initialization for all the methods.

**Results.** As shown in Table 4, SOTA imitation learning method like GAIL and SAILs total fails in this task. Our method GAIL-USN outperforms all the baselines in the Atari game. RIL_CO also performs closely to the optimal performance GAIL*. However, it requires more computation cost due to the usage of co-training. Table 5 shows the performance of GAIL-USN and other baselines on Seaquest game across multiple *state-dependent* action noise.

Table 4: Average return of GAIL, GAIL-USN, RIL, RIL_CO, SAIL-soft and SAIL-hard on the Atari game with 30% *state-dependent action noise* in both the demonstration and the environment.

|  | GAIL* | GAIL-USN (Ours) | GAIL | RIL | RIL_CO | SAIL-soft | SAIL-hard |
|---|---|---|---|---|---|---|---|
| Seaquest | 1,553.5 | **1,564.7** | 159.4 | 776.3 | 1,506.6 | 29.7 | 43.2 |

## F EXTENSION TO GENERAL Q-LEARNING METHOD

USN is a general method for robust training of imitation learning against action noise. BCQ-USN can easily apply USN on top of the generative model of BCQ methods. Because the generative model is similar to a classifier in label-noise learning tasks. Therefore, BCQ-USN can naturally employ the noise-tolerant methods for positive learning and negative learning. However, many IL methods

---

[1]https://github.com/naivety77/gail_atari

Table 5: Average return of our GAIL-USN and other SOTA baselines on the Seaquest game with diverse levels of *state-dependent action noise* in the demonstrations and environments. The reported results are average across 10 random seeds.

| Method | Noise Rates | | | |
| --- | --- | --- | --- | --- |
| | 0.1 | 0.2 | 0.3 | 0.4 |
| GAIL* | 1,553.5 | 1,553.5 | 1,553.5 | 1,553.5 |
| GAIL | **1,695.7** | 1,321.3 | 159.4 | 1,214.6 |
| GAIL-USN | 1,648.6 | **1,677.6** | **1,564.7** | 713.4 |
| RIL | 454.0 | 396.0 | 776.3 | 761.9 |
| RIL_CO | 1,303.0 | 993.3 | 1,506.6 | **1,538.0** |
| SAIL-soft | 0.0 | 760.6 | 29.7 | 455.7 |
| SAIL-hard | 0.0 | 760.6 | 43.2 | 455.7 |

do not have such generative model. How to extend USN to such IL methods is an interesting and essential problem. In this section, we show how to apply USN on some general Q-learning based IL methods, e.g. the conservative Q-learning (CQL) (Kumar et al., 2020).

## F.1 CONSERVATIVE Q-LEARNING (CQL)

Offline RL algorithms promise to learn effective policies from previously-collected, static datasets without further interaction. However, in practice, offline RL presents a major challenge, and standard off-policy RL methods can fail due to overestimation of values induced by the distributional shift between the dataset and the learned policy, especially when training on complex and multi-modal data distributions. Kumar et al. (2020) proposed conservative Q-learning (CQL), which aims to address these limitations by learning a conservative Q-function such that the expected value of a policy under this Q-function lower-bounds its true value. They theoretically show that CQL produces a lower bound on the value of the current policy and that it can be incorporated into a policy learning procedure with theoretical improvement guarantees. In practice, CQL augments the standard Bellman error objective with a simple Q-value regularizer which is straightforward to implement on top of existing deep Q-learning and actor-critic implementations. Eq. 11 addresses the impact of out-of-distribution actions and obtain conservative value estimates. Algorithm 5 presents the procedures of CQL. To use CQL for offline imitation learning, we train an intrinsic curiosity module (ICM) (Pathak et al., 2017) for generating intrinsic rewards.

$$\min_Q \alpha \mathbb{E}_{s \sim \mathcal{D}} \left[ \log \sum_a \exp(Q(s,a)) - \mathbb{E}_{a \sim \hat{\pi}_\beta(a|s)}[Q(s,a)] \right] + \frac{1}{2} \mathbb{E}_{s,a,s' \sim \mathcal{D}} \left[ \left( Q - \hat{\mathcal{B}}^{\pi_k} \hat{Q}^k \right)^2 \right] \quad (11)$$

where $\hat{\mathcal{B}}^{\pi_k} \hat{Q}^k(s,a) = r(s,a) + \gamma \mathbb{E}_{s' \sim P(s'|s,a)}[\max_{a'} \hat{Q}(s',a')]$ is the Bellman optimality operator.

---

**Algorithm 5** CQL

1: Initialize Q-function $Q_\theta$, and optionally a policy $\pi_\phi$.
2: **for** $t = 1, \cdots, T$ **do**
3:     Train the Q-function using objective from Eq. (11).
4:     (only with actor-critic) Improve policy $\pi_\phi$ via with SAC-style entropy regularization.
5: **end for**

---

## F.2 CQL-USN

Since there is no generative model in CQL, we apply USN on top of the Q network. We first introduce a robust IL baseline algorithm, CQL-GCE, by training CQL with additional GCE loss on Q:

$$\mathcal{L}_{pos} = \frac{(1 - Q(s,a)^q)}{q}, \quad (12)$$

We use CQL-GCE for positive learning on the full-batch data. Then, we estimate the predictive uncertainty measures (calibration errors) on Q. Next, we set the sample-selection threshold using the calibration errors, and select large-loss samples for soft negative learning. This simple extension is called CQL-USN. In CQL-USN (Algorithm 6), the Q network is trained in a multi-task manner, predicting values and noise-tolerant action prediction.

---

**Algorithm 6** Robust CQL (e.g. CQL-USN)

---

1: Initialize Q-function $Q_\theta$, and optionally a policy $\pi_\phi$.
2: **for** $t = 1, \cdots, T$ **do**
3:   Update Q-function parameters by minimizing Eq. (11) and USN in Algorithm 1.
4:   (only with actor-critic) Improve policy $\pi_\phi$ via with SAC-style entropy regularization.
5: **end for**

---

### F.3 EXPERIMENTS AND RESULTS

**Settings.** We evaluate the effectiveness of CQL-USN on offline datasets of Atari games from the d4rl benchmark (Fu et al., 2020). We build our method on top of d3rlpy library (Takuma Seno, 2021), and use its default hyperparameter settings.

Table 6: Average return of CQL, CQL-GCE and CQL-USN on Q*bert with *state-dependent action noise*.

| Noise rate | 0.1 |
|---|---|
| CQL | 4,911.0 |
| CQL-GCE | 5,088.5 |
| CQL-USN | **5,455.0** |

Table 7: Average return of CQL, CQL-GCE and CQL-USN on Q*bert with symmetric action noise.

| Noise rate | 0.1 | 0.2 | 0.3 | 0.4 |
|---|---|---|---|---|
| CQL | 6,096.0 | 5,050.0 | 4,000.0 | 2,721.5 |
| CQL-GCE | 6,312.5 | 5,434.5 | **4,189.0** | 4,409.0 |
| CQL-USN | **7,752.0** | **5,615.5** | 2,975.0 | **4,933.0** |

**Results.** We compare the average return of CQL, CQL-GCE and CQL-USN with *state-dependent action noise* (Table 6), symmetric action noise (Table 7) and pairflip action noise (Table 8). CQL-USN usually outperforms CQL-GCE and CQL on the three types of action noise significantly across multiple noise rates.

## G MORE PREDICTIVE UNCERTAINTY DETAILS

**Expected Calibration Error (ECE)** (Naeini et al., 2015; Guo et al., 2017) is widely used to measure the predictive uncertainty (model calibration) of a deep network. To approximate the calibration error in expectation, ECE discretizes the probability interval into a fixed number of bins, and assigns each predicted probability to the bin that encompasses it. The calibration error is the difference between the fraction of predictions in the bin that are correct (accuracy) and the mean of the probabilities in the bin (confidence). Intuitively, the accuracy estimates $\mathbb{P}(Y = y | \hat{p} = p)$, and the average confidence is a setting of $p$. ECE computes a weighted average of this error across bins:

$$\text{ECE} = \sum_{b=1}^{B} \frac{n_b}{N} |acc(b) - conf(b)|, \tag{13}$$

where $n_b$ is the number of predictions in bin $b$, $N$ is the total number of data points, and $acc(b)$ and $conf(b)$ are the accuracy and confidence of bin $b$, respectively. ECE as framed in (Naeini et al., 2015) leaves ambiguity in both its binning implementation and how to compute calibration

Table 8: Average return of CQL, CQL-GCE and CQL-USN on Q*bert with pairflip action noise.

| Noise rate | 0.1 | 0.2 |
|---|---|---|
| CQL | 5,830.0 | 4,438.0 |
| CQL-GCE | 5,352.5 | 4,647.5 |
| CQL-USN | **7,130.5** | **5,348.0** |

for multiple classes. In Guo et al. (2017), they bin the probability interval [0; 1] into equally spaced subintervals, and they take the maximum probability output for each datapoint (i.e., the predicted class's probability). We use this for our ECE implementation.

**Adaptive Calibration Error (ACE)** improves ECE by using adaptive calibration ranges. The motivation is that in order to get the best estimate of the overall calibration error the metric should focus on the regions where the predictions are made (and focus less on regions with few predictions) (Nixon et al., 2019). ACE uses an adaptive scheme which spaces the bin intervals so that each contains an equal number of predictions. In detail, ACE takes as input the predictions $P$ (usually out of a softmax), correct labels, and a number of ranges $R$:

$$\text{ACE} = \frac{1}{KR} \sum_{k=1}^{K} \sum_{r=1}^{R} |acc(r,k) - conf(r,k)|, \tag{14}$$

where $acc(r,k)$ and $conf(r,k)$ are the accuracy and confidence of adaptive calibration range $r$ for class label $k$, respectively. **Thresholded Adaptive Calibration Error (TACE)** is identical to ACE, with the only difference being that TACE is only evaluated on values above $\epsilon$.

**Static Calibration Error (SCE)** is a simple extension of Expected Calibration Error to every probability in the multiclass setting. SCE bins predictions separately for each class probability, computes the calibration error within the bin, and averages across bins (Nixon et al., 2019):

$$\text{SCE} = \frac{1}{K} \sum_{k=1}^{K} \sum_{b=1}^{B} \frac{n_{bk}}{N} |acc(b,k) - conf(b,k)|, \tag{15}$$

where $acc(b,k)$ and $conf(b,k)$ are the accuracy and confidence of bin $b$ for class label $k$, respectively; $n_{bk}$ is the number of predictions in bin $b$ for class label $k$; and $N$ is the total number of data points. Unlike ECE, assuming infinite data and infinite bins, SCE is guaranteed to be zero if and only if the model is calibrated.

**Root Mean Square Calibration Error (RMSCE)** (Hendrycks et al., 2018) measures the square root of the expected squared difference between confidence and accuracy at a confidence level. The RMSCE is estimated with the numerically stable formula:

$$\text{RMSCE} = \sqrt{\sum_{b=1}^{B} \frac{n_b}{N} \Big(acc(b) - conf(b)\Big)^2}. \tag{16}$$

## H ABLATION STUDIES

In this subsection, we study the effects of several essential hyper-parameters and components in LunarLander-2 task with *state-dependent action noise* ($\epsilon = 0.45$). This ablation follows the rule of 'study one, fix the others'. The ablation study results are illustrated in Figure 37 and Figure 35.

**Effect of negative learning loss.** We evaluate the performance of our method using the classic complementary-label learning approach, pairwise-comparison (PC) loss (Ishida et al., 2017) for negative learning. We denote this instance as BCQ-USN-PC. Figure 10(c) and Figure 10 show that BCQ-USN-PC is comparable to the original BCQ-USN when learning with *state-dependent action noise*, symmetric action noise and pairflip action noise. This implies that our method can generate many variants by replacing the negative learning loss.

**Effect of sample-selection threshold.** Figure 37 shows that BCQ-USN outperforms BCQ-GCE significantly with diverse calibration error metrics when $\lambda_{neg} = 1.0$. We can set sample-selection

threshold $\tau_u$ as any types of calibration error metric, which demonstrates the good scalability of our method.

**Effect of unlikely action threshold $\tau$ of generative model $G$.** Figure 35(a) shows that BCQ-USN outperforms BCQ-GCE significantly with five calibration error metrics and one fixed value.

**Effect of the weight of negative learning loss $\lambda_{neg}$.** We select $\lambda_{neg}$ from [0.01, 0.1, 1.0], and find our method BCQ-USN achieves the best performance with $\lambda_{neg} = 1.0$, and all three instances outperform BCQ-GCE (the blue dash dot line in Figure 35(b)). **Effect of smooth rate $\alpha$.** To avoid tuning of the smooth rate in soft negative learning loss, we set the smooth rate as the estimated calibration errors. Figure 35(c) shows that ECE and TACE are the worst, and the other four instances all outperform BCQ-GCE with a large margin. The above ablations show that our method is not sensitive to the choices of hyper-parameters. More importantly, the calibration error metrics can be of multiple use, which maintains the robustness of BCQ-USN while avoiding the effort of tuning hyper-parameters.

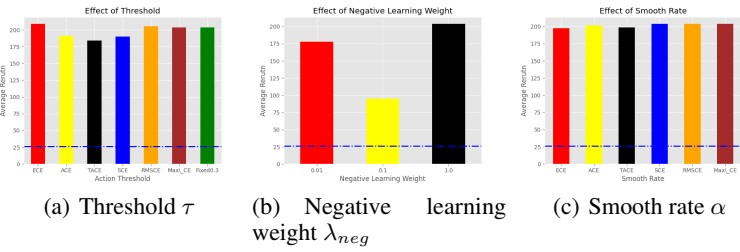

(a) Threshold $\tau$     (b) Negative learning weight $\lambda_{neg}$     (c) Smooth rate $\alpha$

Figure 35: The effects of action threshold $\tau$, smooth rate $\alpha$, and negative learning weight $\lambda_{neg}$.

## H.1 DETAILED ABLATION STUDY RESULTS

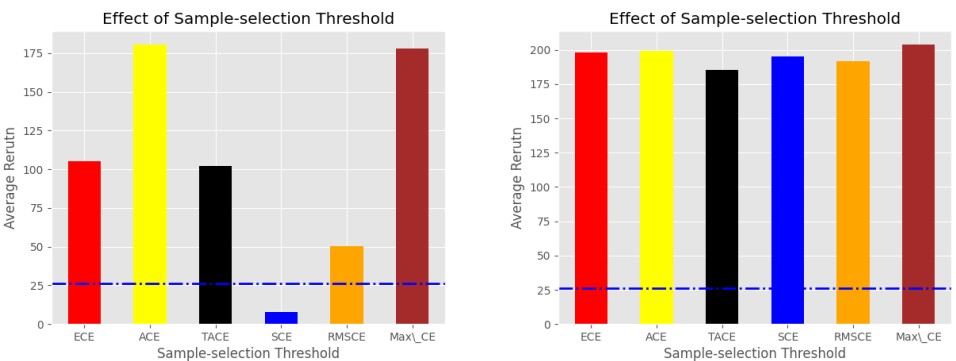

Figure 36: The effect of diverse sample-selection threshold $\tau_u$ when $\lambda_{neg} = 0.01$.

Figure 37: The effect of diverse sample-selection threshold $\tau_u$ when $\lambda_{neg} = 1.0$.

**Effect of sample-selection threshold.** Table 9 and Table 10 shows the ablation results on diverse sample-selection threshold when $\lambda_{neg} = 0.01$ and $\lambda_{neg} = 1.0$, respectively. BCQ-USN with $\lambda_{neg} = 1.0$ outperforms that trained with $\lambda_{neg} = 0.01$ no mater the choice of sample-selection threshold. The corresponding bar chats comparison are shown in Figure 36. BCQ-USN usually outperforms the BCQ-GCE by a large margin when using the estimated calibration error metrics (predictive uncertainty) as the small-loss sample-selection threshold. Therefore, predictive uncertainty is a good alternative to the manual tuning sample-selection threshold, since it requires no prior knowledge about noise and it has good adapbility to diverse noise types and noise rates.

**Effect of threshold $\tau$.** The ablation results on the unlikely action threshold $\tau$ are shown in Table 11. BCQ-USN significantly outperforms BCQ-GCE when setting $\tau$ as different predictive uncertainty

Table 9: Average return of our method on LunarLander-v2 with noise rate $\epsilon = 0.45$, action threshold $\tau = 0.3$ and negative learning weight $\lambda_{neg} = 0.01$ across different small-loss sample-selection threshold.

| Small-loss sample-selection threshold | ECE | ACE | TACE | SCE | RMSCE | Max_CE |
|---|---|---|---|---|---|---|
| LunarLander-v2 | 105.0 | **180.5** | 102.3 | 8.0 | 50.5 | 177.9 |

Table 10: Average return of our method on LunarLander-v2 with noise rate $\epsilon = 0.45$, action threshold $\tau = 0.3$ and negative learning weight $\lambda_{neg} = 1.0$ across different small-loss sample-selection threshold.

| Small-loss sample-selection threshold | ECE | ACE | TACE | SCE | RMSCE | Max_CE |
|---|---|---|---|---|---|---|
| LunarLander-v2 | 198.1 | 199.5 | 185.1 | 195.4 | 191.6 | **203.8** |

measures except ECE. This means that the predictive uncertainty measures are usually reliable choices for eliminating the unlikely action for policy training in BCQ and its variants.

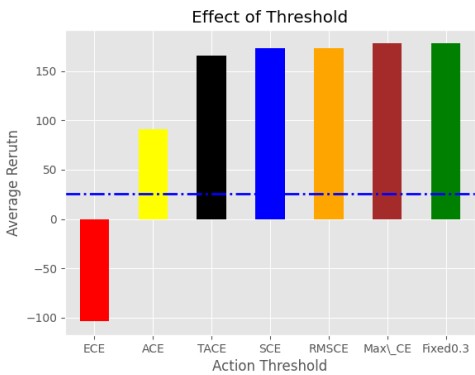

Figure 38: The effect of diverse action threshold $\tau$ when $\lambda_{neg} = 0.01$.

Table 11: Average return of our method on LunarLander-v2 with noise rate $\epsilon = 0.45$ across different threshold $\tau$ of BCQ when $\lambda_{neg} = 0.01$.

| $\tau$ | ECE | ACE | TACE | SCE | RMSCE | Max_CE | Fixed 0.3 |
|---|---|---|---|---|---|---|---|
| LunarLander-v2 | -103.9 | 91.2 | 165.9 | 173.0 | 173.0 | **177.9** | **177.9** |

Table 12: Average return of our method on LunarLander-v2 with noise rate $\epsilon = 0.45$ across different threshold $\tau$ of BCQ when $\lambda_{neg} = 1.0$.

| $\tau$ | ECE | ACE | TACE | SCE | RMSCE | Max_CE | Fixed 0.3 |
|---|---|---|---|---|---|---|---|
| LunarLander-v2 | **208.9** | 191.2 | 184.2 | 189.9 | 205.5 | 203.8 | 203.8 |

**Effect of negative learning weight $\lambda_{neg}$.** Table 13 shows the ablation results on the multiple choices of negative learning loss weight $\lambda_{neg}$.

**Effect of smooth rate $\alpha$.** Table 14 shows the ablation results on using different calibration error metrics as the smooth rate $\alpha$ in soft negative learning loss.

**Effect of negative learning loss.** we compare the performance of the original BCQ-USN with the one using a classic complementary label learning method, e.g. Pairwise-comparison (PC) loss (Ishida et al., 2017) for negative learning:

Table 13: Average return of our method on LunarLander-v2 with noise rate $\epsilon = 0.45$, action threshold $\tau = 0.3$ and smooth rate $\alpha = \text{Max\_CE}$ across different negative learning weight $\lambda_{neg}$.

| $\lambda_{neg}$ | 0.01 | 0.1 | 1.0 |
|---|---|---|---|
| LunarLander-v2 | 177.9 | 94.9 | **203.8** |

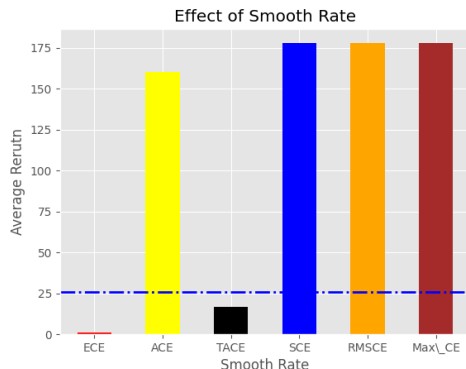

Figure 39: The effect of diverse smooth rate $\alpha$ when $\lambda_{neg} = 0.01$.

Table 14: Average return of our method on LunarLander-v2 with noise rate $\epsilon = 0.45$, action threshold $\tau = 0.3$ and negative learning weight $\lambda_{neg} = 0.01$ across different smooth rate $\alpha$.

| Smooth Rate $\alpha$ | ECE | ACE | TACE | SCE | RMSCE | Max_CE |
|---|---|---|---|---|---|---|
| LunarLander-v2 | 0.9 | 160.4 | 17.0 | 177.9 | 177.9 | **177.9** |

Table 15: Average return of our method on LunarLander-v2 with noise rate $\epsilon = 0.45$, action threshold $\tau = 0.3$ and negative learning weight $\lambda_{neg} = 1.0$ across different smooth rate $\alpha$.

| Smooth Rate $\alpha$ | ECE | ACE | TACE | SCE | RMSCE | Max_CE |
|---|---|---|---|---|---|---|
| LunarLander-v2 | 197.5 | 201.4 | 198.6 | 203.8 | 203.8 | **203.8** |

Ishida et al. (2017) first formulated the problem of complementary-label learning (CLL) and proposed a risk minimization framework. The goal of CLL is to learn a classifier $f$ that minimizes the classification risk (17), from only the complementary labeled samples $\{(x_i, \bar{y}_i)\}_{i=1}^n$. $\{(x_i, \bar{y}_i)\}_{i=1}^n$ are drawn independently from an unknown probability distribution with density: $\bar{p}(x, \bar{y}) = \frac{1}{K-1} \sum_{y \neq \bar{y}} p(x, \bar{y})$, where $K$ is the number of classes.

$$R(f) = \mathbb{E}_{p(x,y)}[\mathcal{L}(f(x), y)]. \tag{17}$$

To achieve this goal, Ishida et al. (2017) proved that the risk (17) can be approximated in an unbiased fashion:

$$\hat{R}(f) = \frac{K-1}{n} \sum_{i=1}^n \bar{\mathcal{L}}\big(f(x_i, \bar{y}_i)\big) - M_1 + M_2. \tag{18}$$

where $\bar{\mathcal{L}}$ is a complementary loss. They considered the one-versus-all (OVA) loss $\bar{\mathcal{L}}_{OVA}(f(x), \bar{y}) = \frac{1}{K-1} \sum_{y \neq \bar{y}} \ell(g_y(x)) + \ell(-g_{\bar{y}}(x))$ and pairwise comparison (PC) loss:

$$\bar{\mathcal{L}}_{PC}(f(x), \bar{y}) = \sum_{y \neq \bar{y}} \ell(g_y(x) - g_{\bar{y}}(x)), \tag{19}$$

where $\ell(z)$ is a binary loss which satisfy $\ell(z) + \ell(-z) = 1$, such as the sigmoid loss $\ell_S(z) = \frac{1}{1+e^z}$. $\bar{\mathcal{L}}_{OVA}$ satisfies this condition with $M_1 = K$ and $M_2 = 2$, and $\bar{\mathcal{L}}_{PC}$ satisfies the condition with $M_1 = K(K-1)/2$ and $M_2 = K - 1$.

Table 16: Average return of offline imitation learning on the LunarLander-v2 with diverse levels of symmetric action noise.

| | Noise Rates | | | | |
|---|---|---|---|---|---|
| Method | 0.05 | 0.25 | 0.45 | 0.65 | 0.85 |
| BCQ | **207.7** | 132.1 | 202.1 | -217.6 | -153.8 |
| BCQ-GCE | 42.3 | 181.9 | 90.8 | 145.0 | -228.8 |
| BCQ-USN-PC | 194.2 | 179.9 | 198.5 | **195.4** | **-119.7** |
| BCQ-USN | 168.8 | **191.2** | **211.1** | 186.0 | -184.7 |

**Results on *state-independent action noise*.** In Table 16 and Table 17, we show the comparative results of BCQ, BCQ-GCE, BCQ-USN and BCQ-USN-PC on LunarLander-v2 with symmetric action noise and pairflip action noise. BCQ-USN is comparable to BCQ-USN-PC, and are usually better than BCQ-GCE and BCQ across diverse noise rates.

Table 17: Average return of offline imitation learning on the LunarLander-v2 with diverse levels of pairflip action noise.

| | Noise Rates | | | | |
|---|---|---|---|---|---|
| Method | 0.05 | 0.25 | 0.45 | 0.65 | 0.85 |
| BCQ | **204.3** | 186.2 | 142.5 | -370.1 | **-122.9** |
| BCQ-GCE | 134.0 | 120.4 | 176.8 | **186.5** | -747.0 |
| BCQ-USN-PC | 200.9 | 200.4 | 186.3 | 172.5 | -799.3 |
| BCQ-USN | 193.0 | **201.2** | **194.5** | 168.0 | -773.8 |

**Results on *state-dependent action noise*.** Training with PC loss, BCQ-USN-PC achieves comparable results to the original BCQ-USN in LunarLander-v2 across multiple levels of *state-dependent action noise* (Table 18).

Table 18: Average return of offline imitation learning on the LunarLander-v2 with diverse levels of *state-dependent action noise*.

| | Noise Rates | | | | |
|---|---|---|---|---|---|
| Method | 0.05 | 0.25 | 0.45 | 0.65 | 0.85 |
| BCQ | **208.0** | **207.4** | -335.2 | -624.3 | -560.2 |
| BCQ-GCE | 172.3 | 188.7 | -159.0 | 67.4 | -518.0 |
| BCQ-USN-PC | 188.3 | 175.8 | **199.0** | -34.8 | -583.2 |
| BCQ-USN | 188.0 | 179.2 | 177.9 | **165.5** | -546.2 |

