# OpenReview forum: "How does Uncertainty-aware Sample-selection Help Decision against Action Noise?"
_ICLR.cc/2023/Conference — Submitted to ICLR 2023_

### Official Review · Reviewer_c6qo · 2022-10-25

**Confidence:** 2
**Correctness:** 2
**Technical Novelty And Significance:** 3
**Empirical Novelty And Significance:** 2
**Recommendation:** 3

**Clarity, Quality, Novelty And Reproducibility:**

Clarification :
1. The authors mention “... where a corrupted label is randomly flipping from other classes…” for state independent noise. Why does state independent noise entail that the action choice is random? It could very well not be random and instead be dependent on exogenous variables not part of the state. Why is this assumption valid? Moreover is this the reason why Algorithm 2 Step 4 creates a complementary batch via random samples?

2. Other concerns raised previously.

Quality, Novelty and Reproducibility :
1. I find the use of negative learning in imitation learning to be novel. (Even though the threshold mechanism seems a direct derivation from one of the predictive uncertainty works.)

2. With the mentioned text and supplementary material, I feel confident that the results can be reproducible.


**Strength And Weaknesses:**

Strengths :
1. The paper seems to be generally well written, where enough background information was provided for good readability.

2. The paper addresses the problem of accounting for (by being robust to) action labelling noise - which in some sense allows for more amateur labellers - which is a welcome direction [however more on this in the weaknesses].

3. I also appreciate the authors’ attempt to showcase why their intuition behind using predictive uncertainty as a means to realize samples on which they must perform negative learning, makes sense via experiments on  NameThisGame.

Weakness :

1. One of my major concerns is the fact that the experiments show the marginal improvements of USN algorithm over the baselines at high levels of noise. Since the labellers are actually humans (in the idealized setup given by the authors), I do not see how experts or amateur labellers have noise as high as 0.5 or even more (basically they are incorrectly labeling half the time?). Even if the work improves the performance measures beyond 0.5 action noise value, does it actually solve the original problem?

2, Secondly, I find that the experimentation setup is very limited and can benefit from testing across more environment setups. Why has different environments been used for different setups of Robustness to state independent noise (Atari not used?, only Lunar Lander), Robustness to state dependent noise (Atari + Lunar)?(& Section 5.1 uses Q*bert)?

3. The authors provide examples of how amateurs or even experts can end up providing noisy action labels. Firstly, I would have thought that it wouldn’t be as high as 0.5, unless authors performed a user study to validate that such high levels are actually seen. Secondly, even if they are seen I would image that it is maybe because of the large action space like continuous action spaces where getting the exact action maybe hard for the labeller. But the environments chosen for the experiments, especially Atari domains mostly have very limited number of actions. For example, only relevant actions in Assault would be left, right, fire, no-op. I do not see why such settings correctly represent authors original motivation from Section 1 and 2.

4. Although I welcome the discussion in section 3.2, I find that the inferences drawn like the correlation between generative model’s loss vs the predictive uncertainty to be quite ambitious from the perspective of the extent of experimentation. I may agree that the said correlation may exist, but, in my view, experiments on a single domain shouldn’t be used to draw conclusions.


**Summary Of The Paper:**

The work proposes an algorithm, USN (uncertainty aware sample selection for Negative learning) for improving performance results in imitation learning, especially under the presence of high action labelling noise. The work seems to be of the class where the demonstration has to be parsed by some human labor whose errors made on action labels are termed as action noise. The key idea is to figure out data samples from the demonstration which have a high predictive uncertainty (which they show is negatively correlated to generative loss), and that negative learning on such samples can help with imitation learning.

**Summary Of The Review:**

I find that the question being investigated by the authors is interesting, which is to accommodate for action noise for Imitation Learning setups, however through their results it seems that the major benefits of this approach are reaped beyond a high enough noise level which I do not believe are typically achieved by amateaur or expert labellers. Among other concerns about the setup, I feel that the work would benefit from a more thorough and consistent experimentation to bolster their claims.

---

> ### Author Response · Authors · 2022-11-19
> **Reponses to Reviewer c6qo**
>
> *Q1*: One of my major concerns is the fact that the experiments show the marginal improvements of USN algorithm mover the baselines at high levels of noise. Since the labellers are actually humans (in the idealized setup given by the authors), I do not see how experts or amateur labellers have noise as high as 0.5 or even more (basically they are incorrectly labeling half the time?). Even if the work improves the performance measures beyond 0.5action noise value, does it actually solve the original problem?
>
> *A1*: Thank you for your suggestions. For the newly added experiments of BC and online imitation learning, we evaluated the robustness and generalization of our method USN on state-independent and state-dependent action noise with noise rates of [0.1, 0.3, 0.5]. The results in Section 5 shows that our method USN can improves the robustness against diverse action noise in the demonstrations.
>
> In the Appendix D.1, we have shown quantitative results on offline imitation learning tasks. Even when the noise rate is as high as 0.65, our method BCQ-USN achieves the highest performance of 167.1, outperforming other baselines BCQ-GCE (113.0) and BCQ (-626.8).
>
> *Q2*: Secondly, I find that the experimentation setup is very limited and can benefit from testing across more environment setups. Why has different environments been used for different setups of Robustness to state-independent noise (Atari not used?, only Lunar Lander), Robustness to state-dependent noise (Atari + Lunar)?(&Section 5.1 uses Q*bert)?
>
> *A2*: Thank you for your constructive comments, we have added more experiments in the revised version. First, we have provided a comprehensive study on the correlations between loss, uncertainty versus noise rates for Behavioral Cloning, online imitation learning and offline imitation learning. For BC model, we evaluate our method USN on two classic control tasks Acrobot-v1 and LunarLander-v2 with both state-independent action noise and state-dependent action noise. For online imitation learning, we show how to improve the robustness of GAIL against state-dependent action noise on Seaquest, Q*bert and Hero games. Due to the limits of time and accessible computation resources during rebuttal, we will add more results for offline imitation learning in the near future.
>
> *Q3*: The authors provide examples of how amateurs or even experts can end up providing noisy action labels. Firstly, I would have thought that it wouldn’t be as high as 0.5, unless authors performed a user study to validate that such high levels are actually seen. Secondly, even if they are seen I would image that it is maybe because of the large action space like continuous action spaces where getting the exact action maybe hard for the labeller. But the environments chosen for the experiments, especially Atari domains mostly have very limited number of actions. For example, only relevant actions in Assault would be left, right, fi re, no-op. I do not see why such settings correctly represent authors original motivation from Section 1 and 2.
>
> *A3*: Thank you for your constructive suggestions. In section 5, we have added new experiments to show how our method can improve the robustness of BC and GAIL against action noise in the demonstrations. In the new experiments, we show our method outperforms the other baselines in classic control tasks (Acrobot-v1 and LunarLander-v2) and Atari games (Seaquest, Q*bert and Hero) across multiple noise rates (0.1, 0.3 and 0.5). We will investigate the continuous action space in the near future.
>
> *Q4*: The work would benefit from a more thorough and consistent experimentation to bolster their claims.
>
> *A4*: Thank you for your constructive suggestions. We have added more experiments in Section 5, to show how our method USN can improves the robustness of BC and online imitation learning algorithm GAIL against both state-independent action noise and state-dependent action noise in the demonstrations.

---

> > ### Author Response · Authors · 2022-11-19
> > **Reponses to Reviewer c6qo continue**
> >
> > *Q5*:  I may agree that the said correlation may exist, but, in my view, experiments on a single domain shouldn’t be used to draw conclusions.
> >
> > *A5*: Thanks for your constructive suggestions. We have provided a comprehensive study on the correlations between loss, uncertainty estimates versus noise rates in Section 3 and Appendix C for Behavioral Cloning, online imitation learning (GAIL) and offline imitation learning (BCQ with ICM rewards).
> >
> > From Figure 3, Figure 4, Figure 5, Figure 6 and more results in the Appendix C, we can conclude that the loss estimation can be used as a criterion for detecting noisy actions using the BC model, BCQ’s generative model, GAIL actor and discriminator. Specifically, the ‘large-loss’ samples have high probability to contain noisy actions. For the BC model, GAIL actor and GAIL discriminator, the uncertainty estimation ECE can also be used as a good criterion for detecting noisy actions. However, for the offline imitation learning method BCQ, ECE does not show a clear positive correlation with the increase of action noise. Instead, the dynamics of ECE in BCQ’s generative model has a negative correlation to the loss. In this paper, we focus on proposing a general method that detects noisy actions with high probability and leverage the selected data to improve the robustness of BC, online IL and offline IL against diverse action noise in the demonstrations. To this end, we propose a general paradigm called Uncertainty-aware Sample-selection with Negative learning (USN) for robust training IL models against action noise. USN selects large-loss samples using the uncertainty estimation, e.g. ECE as a threshold.
> >
> > *Q6*: The authors mention “... where a corrupted label is randomly flipping from other classes…” for state-independent noise. Why does state independent noise entail that the action choice is random? It could very well not be random and instead be dependent on exogenous variables not part of the state. Why is this assumption valid? Moreover, is this the reason why Algorithm 2 Step 4 creates a complementary batch via random samples?
> >
> > *A6*: Thanks for your comments. State-independent action noise comes when an amateur annotator randomly picks an action for some unseen states. The noise generation process is quite similar to the class-conditional label noise, where a corrupted label is randomly flipping from other classes (Symmetric noise) or its neighbor (Pairflip noise). We choose such action noise modeling because the generation procedure is independent of the state contents. We highly agree with your opinion that it could very well not be random and instead be dependent on exogenous variables not part of the state. We will consider such action noise modeling in the future. Thank you for your suggestion.
> >
> > As clarified in Section 4, our main goal is to develop a general robust IL paradigm against diverse type of action noise. To achieve this goal, we design USN to composite of two main steps: uncertainty-aware sample-selection and negative learning for loss correction.
> > Uncertainty-aware sample-selection aims to select samples that contains noisy actions with high probability. Given a mini-batch of demonstration data with a size of B, we first employ a basic IL method with parameter ω, for positive learning on the full-batch data. Then in second step, we estimate the uncertainty, e.g. on the batch data and regards it as a threshold τu for selecting large-loss samples. Next, we sample the large-loss batch  ̃M by selecting the large-loss samples with a length of: #neg = (1 − τu) ∗ B.
> >
> > Negative learning for loss correction. Positive learning with full-batch data with noisy actions will results in bias in the loss training, and misguides the policy to choose the wrong actions. To correct the loss bias, we propose to leverage the selected large-loss sample for negative learning. Intuitively, the selected large-loss batch $\tilde{M}$ contains noisy actions with high probability. Its complementary set has more chances to contain true actions. Therefore, we generate complementary actions for $\tilde{M}$ by randomly selecting $\bar{a}$ from $\{1, .., |\mathcal{A}|\} \backslash \{a\}$, resulting a complementary bath $\bar{M}$ for negative learning.
> > Negative learning on the complementary batch of the selected large-loss samples will correct the loss bias from action noise, and therefore improve the imitation learning performance.
> > To further boost the performance, we implement negative learning with label smoothing, resulting \textit{Soft Negative learning}.
> > Specifically, we employ the following negative log likelihood (NLL) loss for negative learning on the `large-loss' samples with label smoothing.

---

> ### Author Response · Authors · 2022-12-13
> **Follow up**
>
> Dear Reviewer c6qo,
>
> Thank you for your great efforts for reviewing our paper.
>
> We have made many modification to our manuscript to address your major concerns. Would you please read our responses and follow up to update your reviews accordingly. Also, any further constructive comments and suggestions are welcome to further improve our paper. Thank you!

---

### Official Review · Reviewer_k6dx · 2022-10-28

**Confidence:** 4
**Correctness:** 2
**Technical Novelty And Significance:** 1
**Empirical Novelty And Significance:** 2
**Recommendation:** 3

**Clarity, Quality, Novelty And Reproducibility:**

The clarity of this work can be greatly improved; there are several typos that impact the readability of this work; for example the usage of the word “labor” vs. “annotator” in the early parts of the paper, to the description of the various imitation learning algorithms — this work only really uses BCQ and derivatives (offline), so having BC in this section is a bit of a red herring.

Figure 4 also has a method labeled “BCQ-USN-PC” — I think this is supposed to be “BCQ-USN-MC” for MC dropout, but in general, there various differences between methods could be made much more clear (in text, and in the graphs).

**Strength And Weaknesses:**

I think that this paper presents a neat idea; action noise is a huge problem for imitation learning as a whole; however, the proposed approach and evaluation do not help address this, or truly evaluate the generalizability of the USN approach at all.

It’s not clear to me why this work chooses to implement USN on top of BC-Q as the base offline IL algorithm, relative to something like standard Behavioral Cloning (BC). BC-Q requires reward annotations to learn the Q-function, and; while this work correctly avoids using the ground truth environment reward (which would implicitly tell you what actions are/aren’t suboptimal; basically turning this into an offline RL problem), they instead **choose to train an intrinsic reward component following Pathak et. al. 2017** — the form of this intrinsic reward is that of a predictive model that uses surprisal against a learned forward dynamics model as a signal. However, in follow-up work by the same authors (Pathak et. al. 2018), this form of intrinsic reward is **explicitly stated to be problematic in the cases of stochastic environments and crucially, when actions are noisy/suboptimal, as it impacts learning the predictive model. These environments are mostly deterministic, and the noise (when injected) is done so in targeted ways that don’t seem to reflect the real-world settings hinted at in the introduction.

I fundamentally don’t trust these results; there are a lot of design choices that seem poorly motivated (why skip straight to Monte Carlo dropout instead of just using entropy of predictions, given actions are discrete), and I think that for there to be a true claim of addressing problems of action noise in imitation learning, we’d need at least one-two experiments in continuous control settings where this problem is more prevalent and damaging.

**Summary Of The Paper:**

This work presents an approach for learning better policies for imitation learning (both offline/online) in the presence of action noise; notably, this work makes a distinction between state-independent action noise (e.g., an annotator picking a random action for a state) vs. state-dependent action noise (e.g., a complex motion in a narrow part of the state space that affects action labels in a correlated way).

To handle these types of noise, this work proposes uncertainty-aware sample selection with soft negative sampling (USN), a framework that can be plugged into existing imitation learning algorithms like BC-Q; the punchline is to first train an imitation learning policy from all demonstrations, estimate the predictive uncertainty via e.g., Expected Calibration Error (ECE), and use the corresponding threshold to identify “large-loss” examples to inform a negative sampling procedure. The key assumption here is that “large-loss” state-action pairs correspond to transitions with high-uncertainty; rather than “force” a policy to act subject to the given label, it’s better to just replace the given label with a “random” action —> essentially increasing the entropy/model uncertainty at that state, leading to a better policy.

Rather than build the results on a “simple” imitation learning algorithm like behavioral cloning, this work instead builds USN (and corresponding results) on BC-Q, training a separate intrinsic reward component (Pathak et. al., 2017) to act as the reward for learning the Q function.

The results on a few discrete action environments from Atari (Name This Game, Seaquest, Lunar Lander) —> show that USN outperforms BC-Q and online IL approaches as the noise ratio increases (state-independent), and in cases of state-dependent noise.

**Summary Of The Review:**

While the motivation behind the approach is well-formed, I don’t think the proposed algorithm for using large-loss examples as negatives to harden imitation learning policies to action noise is evaluated thoroughly. There are a lot of hard-to-swallow assumptions in the current evaluation (using BC-Q with a learned intrinsic reward component — a method that fundamentally is flawed and doesn’t generalize to more stochastic environments), and the choice of only evaluating discrete action spaces is questionable.

I’d love to see 1) experiments using USN on top of traditional Behavioral Cloning, and 2) experiments in continuous action spaces with real-world noise (e.g., from humans) to truly believe the proposed USN framework generalizes.

---

> ### Author Response · Authors · 2022-11-19
> **Reponses to Reviewer k6dx**
>
> *Q1*: The key assumption here is that “large-loss” state-action pairs correspond to transitions with high-uncertainty; rather than “force” a policy to act subject to the given label, it’s better to just replace the given label with a “random” action —> essentially increasing the entropy/model uncertainty at that state, leading to a better policy.
>
> *A1*: Nice suggestion! Due to limits of time and computational resources in rebuttal, we have provided a comprehensive study on the correlations between loss, uncertainty versus noise rates for BC, GAIL and BCQ. For GAIL, we have studied such correlations for both actor and discriminator, showing the loss and some uncertainty estimates are good indicator for noisy action detection. In the near future, we will investigate your suggestion on more general actor-critic learning. Thanks.
>
> *Q2*: It’s not clear to me why this work chooses to implement USN on top of BC-Q as the base offline IL algorithm, relative to something like standard Behavioral Cloning (BC).
>
> *A2*: Thanks for your suggestions. In the revised paper, we have provided a comprehensive study on the correlations between loss, uncertainty versus noise rates for Behavioral Cloning, online imitation learning and offline imitation learning in Section 3. Correspondingly in the experiments section (Section 5), we have demonstrated how our proposed method USN improves the robustness of BC, offline imitation learning (GAIL) and offline imitation learning (BCQ and CQL with ICM rewards) against diverse action noises in the demonstrations.
>
> In the first experiment (Section 5.2), we evaluate the robustness of our method USN on the classic control tasks - Acrobot-v1 and LunarLander-v2. Figure 7 and Figure 8 show the performance of BC, GCE and our USN under multiple state-independent action noise and state-dependent action noise, respectively. In both types of action noise in the demonstrations, our method USN achieves better performance than other baselines.
>
> Then in Section 5.3, we show how our method USN can improves the robustness of online imitation learning against action noise. We apply our method USN on the BC initialization and compares to SOTA robust imitation learning algorithms, RIL_CO and SAIL with soft weights. The results under multiple state-dependent action noise for three Atari games are shown in Figure 9. SAIL totally fails in all the Atari games under action noise in the demonstrations. In the Seaquest game, both our method GAIL-USN and RIL_CO achieves the best performance; while RIL_CO fails in the other two games. Our method GAIL-USN outperforms all the baselines in the other two Atari games. More details are summarized in Appendix E.
>
> *Q3*: These environments are mostly deterministic, and the noise (when injected) is done so in targeted ways that don’t seem to reflect the real-world settings hinted at in the introduction.
>
> *A3*: Thanks for your constructive comments. Due to the limit of time and accessible computational resources during rebuttal, we have added experiments to show the robustness of our USN method on Behavioral Cloning, online imitation learning and offline imitation learning on classic control tasks (Acrobot-v1 and LunarLander-v2) and several Atari games (Seaquest, Q*bert and Hero). In the near future, we are interested in extending our method to continuous control tasks by using the regression uncertainty estimations, and more challenging real-world tasks.
>
> *Q4*: The clarity of this work can be improved. For example, the usage of the word “labor” vs. “annotator” in the early parts of the paper. This work only really uses BCQ and derivatives (offline), so having BC in this section is a bit of a red herring. Figure 4 also has a method labeled “BCQ-USN-PC” — I think this is supposed to be “BCQ-USN-MC” for MC dropout, but in general, there various differences between methods could be made much more clear (in text, and in the graphs).
> *A4*: Thank you for your constructive comments. We will continuously improve the clarity of our work. We have changed the world “labor” to “annotator” in current version.
>
> To study the effect of negative learning loss. We evaluate the performance of our method using the classic complementary-label learning approach, pairwise-comparison (PC) loss [1] for negative learning. We denote this instance as BCQ-USN-PC. Figure 10 shows that BCQ-USN-PC is comparable to the original BCQ-USN when learning with state-independent action noise (symmetric action noise and pairflip action noise) and state-dependent action noise. This implies that our method can generate many variants by replacing the negative learning loss.
>
> [1] Takashi Ishida, Gang Niu, Weihua Hu, and Masashi Sugiyama. Learning from complementary labels. arXiv preprint arXiv:1705.07541, 2017.

---

> > ### Author Response · Authors · 2022-11-19
> > **Reponses to Reviewer k6dx continue**
> >
> >
> > *Q5*: I’d love to see 1) experiments using USN on top of traditional Behavioral Cloning, and 2) experiments in continuous action spaces with real-world noise (e.g., from humans) to truly believe the proposed USN framework generalizes.
> >
> > *A5*: Due to the limits of time and accessible computations resources during rebuttal, we have added a comprehensive study of the correlations between loss, uncertainty versus noise rates for Behavioral Cloning, online imitation learning and offline imitation learning on Section 3 and Appendix C. In the experiments part (Section 5), we have demonstrated how to use our method USN to improve the robustness of BC, online IL and offline IL against diverse action noise in the demonstrations. In the near future, we are interested in extending our method to continuous control tasks by using the regression uncertainty estimations, and more challenging real-world tasks.

---

> ### Author Response · Authors · 2022-12-13
> **Follow up**
>
> Dear Reviewer k6dx,
>
> Thank you for your great efforts for reviewing our paper.
>
> We have made many modification to our manuscript to address your major concerns. Would you please read our responses and follow up to update your reviews accordingly. Also, any further constructive comments and suggestions are welcome to further improve our paper. Thank you!

---

### Official Review · Reviewer_gQZL · 2022-11-03

**Confidence:** 4
**Correctness:** 3
**Technical Novelty And Significance:** 3
**Empirical Novelty And Significance:** 3
**Recommendation:** 6

**Clarity, Quality, Novelty And Reproducibility:**

The paper is overall well-written and easy to follow. The authors do not include their code in the supplementary, so it is hard to evaluate the reproducibility.

**Strength And Weaknesses:**

Strengths
* Imitation learning from noisy demonstrations is a practical setting.
* The motivation is clear and reasonable.
* Empirical result shows good performance.

Weaknesses
* The performance of the method seems to quite rely on the choice of non-optimal demonstrations. The non-optimal demonstrations are defined as demonstrations with noise in the paper, in such a way the uncertainty estimation technology can be used to find the noisy demonstrations. However, what if we define non-optimal demonstrators as early-stage RL training checkpoints. Can the uncertainty estimation scheme still find the non-optimal demonstrations?
* In Atari domain, the authors only choose 1 game for evaluation. Results on more Atari games can make the results more convincing. Also, more details about the experiments can be given. For example, in Atari domain, how many demonstrations are used for GAIL training? Is there is a need to specially design GAIL since many works point out that directly applying GAIL into Atari domains leads to bad performance.

**Summary Of The Paper:**

This paper investigates a practical setting in imitation learning, where a fraction of expert demonstrations are noisy demonstrations. The authors propose to select hard samples by measuring the uncertainty and update the model with the selected samples. The motivation comes from the neural network, which tends to fit easy samples first. The additional training on the selected hard samples can thus guarantee the generalisation of the model. Empirical results show the effectiveness in Box2D and one Atari games.

**Summary Of The Review:**

Overall, the paper is well-written and easy to follow.  The method is sound and somewhat novel. However, I have concerns about the generalisation of the method since it seems to rely on the choice of non-optimal demonstrations. Also, some implementation details in the experiment seems missing. As a result, I provide my initial score as ``boardline accept''

---

> ### Author Response · Authors · 2022-11-19
> **Responses to Reviewer gQZL**
>
> *Q1*: The performance of the method seems to quite rely on the choice of non-optimal demonstrations. The non-optimal demonstrations are defined as demonstrations with noise in the paper, in such a way the uncertainty estimation technology can be used to find the noisy demonstrations. However, what if we defi ne non-optimal demonstrators as early-stage RL training checkpoints. Can the uncertainty estimation scheme still find the non-optimal demonstrations?
>
> *A1*: Yes. In Section 3 and Appendix C, we have provided a comprehensive study to show the correlation between loss, several uncertainty estimations versus increasing noise rates for Behavioral Cloning, online imitation learning, and offline imitation learning, separately.
>
> For the noisy demonstration with suboptimal demonstrations, we study correlation between loss, uncertainty estimation of GAIL versus optimality of the demonstrations in the Appendix C.2. The results of GAIL actor and discriminator in Q*bert game are shown in Figure 17 and Figure 18. The lower load iter denotes worse optimality. The results show that with the decreasing of the optimality in the demonstrations, the loss estimation of the GAIL actor increases accordingly. For the uncertainty estimations, ECE and MCE show the positive correlation with the loss estimation and the increase of noise rate. Entropy, MaxProb and MC\_dropout show light positive correlations. For the GAIL discriminator, the loss estimation and many uncertainty estimations (i.e. ECE, MCE, Bald, MC\_dropout and NUQ) hold the positive correlation with the decrease of optimality in the demonstrations.
>
> The above observations hold for many Atari games. Therefore, the loss estimation, the uncertainty estimations - ECE, MCE and even MC\_dropout, can be used as criteria for detecting suboptimal data in the demonstrations using the GAIL actor.  Similarly, the loss estimation, the uncertainty estimations - ECE, MCE, Bald, MC\_dropout and NUQ, can be used as criteria for detecting suboptimal data in the demonstrations using the GAIL discriminator.
>
> *Q2*: In Atari domain, the authors only choose 1 game for evaluation. Results on more Atari games can make the results more convincing. Also, more details about the experiments can be given. For example, in Atari domain, how many demonstrations are used for GAIL training? Is there is a need to specially design GAIL since many works point out that directly applying GAIL into Atari domains leads to bad performance.
>
> *A2*: Since most of the previous robust online imitation learning algorithms [1,2] are based on GAIL, we also choose GAIL as our base method. However, the original GAIL can not achieve good performance in high-dimensional environment like Atari games [3,4]. Fortunately, we found that using Behavioral Cloning as an initialization for the actor, GAIL is able to achieve good performance on some Atari games, e.g. Seaquest, Q*bert and Hero with only one full-episode demonstration. Our implementation is based on the opensource code: https://github.com/naivety77/gail_atari
>
> The results of GAIL with BC as an initialization are shown in the following table:
> | Method    | Pong | Seaquest | BeamRider | Hero | Qbert |
> | :---      |:---: | :---:    |  :---:    | :---:| :---: |
> |   BC      | -20.7(0.46) | 200.0(83.43) | 1028.4(396.37) | 7782.5(50.56) | 11420.0(3420.0) |
> | GAIL      |  -1.73(18.1)| 1474.0(201.6)| 1087.6(559.09) | 13942.5(67.13)| 8027.27(24.9)   |
> | GAIL+BC   | 21.0(0.0) | 1662.0(161.85) | 2306.4(1527.23) | 20020(22.91) | 13225.0(1347.22) |
> | PPO(Best) | 21.0(0.0)| 1840(0.0)| 2637.45(1378.23)| 27814.09(46.01) | 15268.18(127.07) |
>
> [1] Voot Tangkaratt, Nontawat Charoenphakdee, and Masashi Sugiyama. Robust imitation learning from noisy demonstrations. arXiv preprint arXiv:2010.10181, 2020.
>
> [2] Yunke Wang, Chang Xu, and Bo Du. Robust adversarial imitation learning via adaptively-selected demonstrations. In IJCAI, pp. 3155–3161, 2021.
>
> [3] Aaron Tucker, Adam Gleave, and Stuart Russell. Inverse reinforcement learning for video games.
> Workshop on Deep Reinforcement Learning at NeurIPS, 2018.
>
> [4] Daniel S Brown, Wonjoon Goo, Prabhat Nagarajan, and Scott Niekum. Extrapolating beyond
> suboptimal demonstrations via inverse reinforcement learning from observations. In International
> Conference on Machine Learning, 2019a.
>
> [5] Daniel S Brown, Wonjoon Goo, and Scott Niekum. Ranking-based reward extrapolation without
> rankings. In Conference on Robot Learning, 2019b.

---

### Author Response · Authors · 2022-12-13
**General responses**

Dear AC and reviewers,

Thank your for your great efforts on reviewing our paper!

In the updated version, we have made important modifications to address the concerns and constuctive suggestions from the reviewers. Specifically,

(1) we have provided a comprehensive study on the correlations between loss and uncertainty across diverse action noise for behavioral cloing, online imitation learning and offline imitation learning.

(2) Based on the consistent correlations, we proposed a general paradigm for robust imitation learning against diverse action noise.

(3) Our method USN is scalable to behavioral cloning, online imitation leanring and offline imitation learning.

Would you please read our responses and update reviews accordingly. We also would like to accept any further constructive comments to help further improve our paper. Thank you!

---

### Decision · Program_Chairs · 2023-01-20

**Decision:**

Reject

**Justification For Why Not Higher Score:**

Major concerns by two of the three reviewers. The positive reviewers agrees with limited empirical evaluation.

**Justification For Why Not Lower Score:**

N/A

**Metareview: Summary, Strengths And Weaknesses:**

I thank the authors for their submission and active engagement during the discussion period. This paper is borderline. While the paper is well written, easy to follow and the problem interesting, all reviewers have remarked the limited empirical results (only one Atari game). I agree with reviewers k6dx  and c6qo concerns around the evaluation. Therefore I recommend rejection but encourage the authors to act upon the reviewers suggestions to improve their work.